# Condensation of Ede1 promotes the initiation of endocytosis

Mateusz Kozak, Marko Kaksonen*

Department of Biochemistry and NCCR Chemical Biology, University of Geneva, Geneva, Switzerland

**Abstract** Clathrin-mediated endocytosis is initiated by a network of weakly interacting proteins through a poorly understood mechanism. Ede1, the yeast homolog of mammalian Eps15, is an early-arriving endocytic protein and a key initiation factor. In the absence of Ede1, most other early endocytic proteins lose their punctate localization and endocytic uptake is decreased. We show that in yeast cells, cytosolic concentration of Ede1 is buffered at a critical level. Excess amounts of Ede1 form large condensates which recruit other endocytic proteins and exhibit properties of phase-separated liquid droplets. We demonstrate that the central region of Ede1, containing a coiled-coil and a prion-like region, is essential for both the condensate formation and the function of Ede1 in endocytosis. The functionality of Ede1 mutants lacking the central region can be partially rescued by an insertion of heterologous prion-like domains. Conversely, fusion of a heterologous lipid-binding domain with the central region of Ede1 can promote clustering into stable plasma membrane domains. We propose that the ability of Ede1 to form condensed networks supports the clustering of early endocytic proteins and promotes the initiation of endocytosis.

## Editor's evaluation

This article demonstrates that the early arriving endocytic protein Ede1, the yeast Eps15 homolog, can generate a separate liquid-phase in vivo when overexpressed and that the domains conferring this property are required and sufficient to nucleate endocytic patches. The genetic and microscopy results are compelling and they support the model of liquid-phase separation in endocytosis, even though alternatives are discussed.

*For correspondence:
marko.kaksonen@unige.ch

**Competing interest:** The authors declare that no competing interests exist.

## Introduction

Clathrin-mediated endocytosis is a process which eukaryotic cells use to produce small transport vesicles from their plasma membrane. These vesicles deliver cargo molecules to the endolysosomal trafficking network where they are sorted for recycling or degradation (*Grant and Donaldson, 2009*). This process is the primary route of internalization of extracellular and surface molecules in eukaryotic cells (*Kirchhausen et al., 2014*).

Clathrin-mediated endocytosis requires a complex protein machinery to assemble on the plasma membrane in a specific sequence (*Kaksonen et al., 2005*; *Sirotkin et al., 2010*; *Taylor et al., 2011*). Endocytosis starts with the arrival of pioneer proteins which select the site and initiate the assembly of the endocytic coat (*Kaksonen and Roux, 2018*). Different pioneer proteins (*Henne et al., 2010*; *Cocucci et al., 2012*; *Ma et al., 2016*), lipids (*Antonescu et al., 2011*) and cargo molecules (*Liu et al., 2010*; *Layton et al., 2011*) have been shown to promote the initiation step (reviewed by *Godlee and Kaksonen, 2013*). However, the exact mechanism by which the endocytic sites are initiated remains poorly understood.

The molecular mechanisms of clathrin-mediated endocytosis have been studied extensively in yeasts, such as the budding yeast *Saccharomyces cerevisiae*. The pioneer module in yeast includes highly conserved adaptor proteins which bind membrane and cargo, such as the adaptor protein complex 2 (AP-2 complex), Syp1 (FCHo1/2 in mammals) and Yap1801/2 (AP180). Two conserved scaffold proteins, clathrin and Ede1 (Eps15), also arrive during the early phase. In contrast to the remarkably well-ordered assembly of the membrane-bending phase (*Picco et al., 2015*), the early proteins lack a specific sequence of recruitment (*Carroll et al., 2012*; *Pedersen et al., 2020*). Curiously, all the genes coding for the earliest-arriving proteins can be deleted without completely blocking endocytosis (*Brach et al., 2014*). However, the frequency of endocytic events is decreased in such cells, and the ability to regulate cargo recruitment is drastically compromised.

Ede1, a homologue of mammalian Eps15, is one of the key early proteins. It is among the earliest to appear at the nascent endocytic site (*Carroll et al., 2012*). The deletion of the EDE1 gene reduces the overall membrane uptake by 35 % (*Gagny et al., 2000*) and decreases the number of productive ndocytic events by 50 % (*Carroll et al., 2012*; *Kaksonen et al., 2005*). Ede1 interacts via its three Eps15 homology (EH) domains and other interaction motifs with proteins such as the AP-2 complex, Hrr25, epsins, Sla2, Syp1, and Yap1801/2 (*Maldonado-Báez et al., 2008*; *Reider et al., 2009*; *Peng et al., 2015*). Several of these proteins depend on interaction with Ede1 to become enriched at the endocytic sites (*Carroll et al., 2012*). Ede1 also oligomerizes via its coiled-coil domain, which is required for it to properly localise and function (*Boeke et al., 2014*; *Lu and Drubin, 2017*).

In recent years, liquid-liquid phase separation of biomolecules has garnered much attention as a mechanism for assembly of cellular organelles which lack surrounding membranes, such as P granules, nucleoli and stress granules (*Banani et al., 2017*). Such membrane-less compartments can accelerate reactions, sequester molecules from the cytoplasm or establish spatial organisation (*Lyon et al., 2021*). The phase separation framework can also apply to sub-micrometre compartments that balance the needs of concentrating select components with allowing a dynamic exchange and rearrangement of molecules. Examples of such compartments include transcriptional superenhancers (*Sabari et al., 2018*) and a wide array of membrane receptor clusters (*Case et al., 2019*).

In this work, we show that Ede1 has a propensity to form cellular condensates. We demonstrate that the cytosolic concentration of Ede1 is buffered at a critical concentration. We identify the molecular features driving the condensation of Ede1 and show that they are essential for the normal function of Ede1 during endocytosis. Our findings suggest that Ede1 has a ability form molecular condensates, which promote the initiation and maturation of endocytic sites. The Ede1 condensates exhibit many of the hallmark properties of phase separated liquids.

## Results

### Ede1 can form dynamic protein condensates

In normal yeast cells, fluorescently tagged Ede1 localizes to endocytic sites at the plasma membrane (*Kukulski et al., 2012*). However, we discovered previously that under certain experimental conditions Ede1 can also assemble into large condensates (*Boeke et al., 2014*). These condensates were seen in cells that either overexpressed Ede1, or expressed Ede1 at normal levels, but lacked three early endocytic adaptors. Although these condensates are abnormal structures that have not been observed in wild-type cells, we reasoned that studying them in more detail might provide insights into the mechanism by which Ede1 promotes the assembly of the early endocytic proteins.

To visualize the condensates, we expressed Ede1 fused to enhanced green fluorescent protein (EGFP) from its endogenous locus in haploid wild-type cells, or in cells from which three endocytosis-related genes were deleted (*yap1801Δ yap1802Δ apl3Δ*, called 3×ΔEA for short). These genes code for early-arriving endocytic adaptor proteins Yap1801, Yap1802, and the α-subunit of the AP-2 complex. Alternatively, we overexpressed EGFP-Ede1 from its endogenous locus under the control of a strong heterologous promoter.

We observed that part of the cellular Ede1-EGFP in the mutant strains localized into condensates that were much brighter than the normal endocytic sites (*Figure 1A*). The condensates in the overexpression strain were larger and brighter than those in the 3×ΔEA cells (*Figure 1B*). The condensates usually associated with the plasma membrane, but were also observed away from it (*Figure 1A*). In contrast, normal endocytic sites are always associated with the plasma membrane. The condensates in

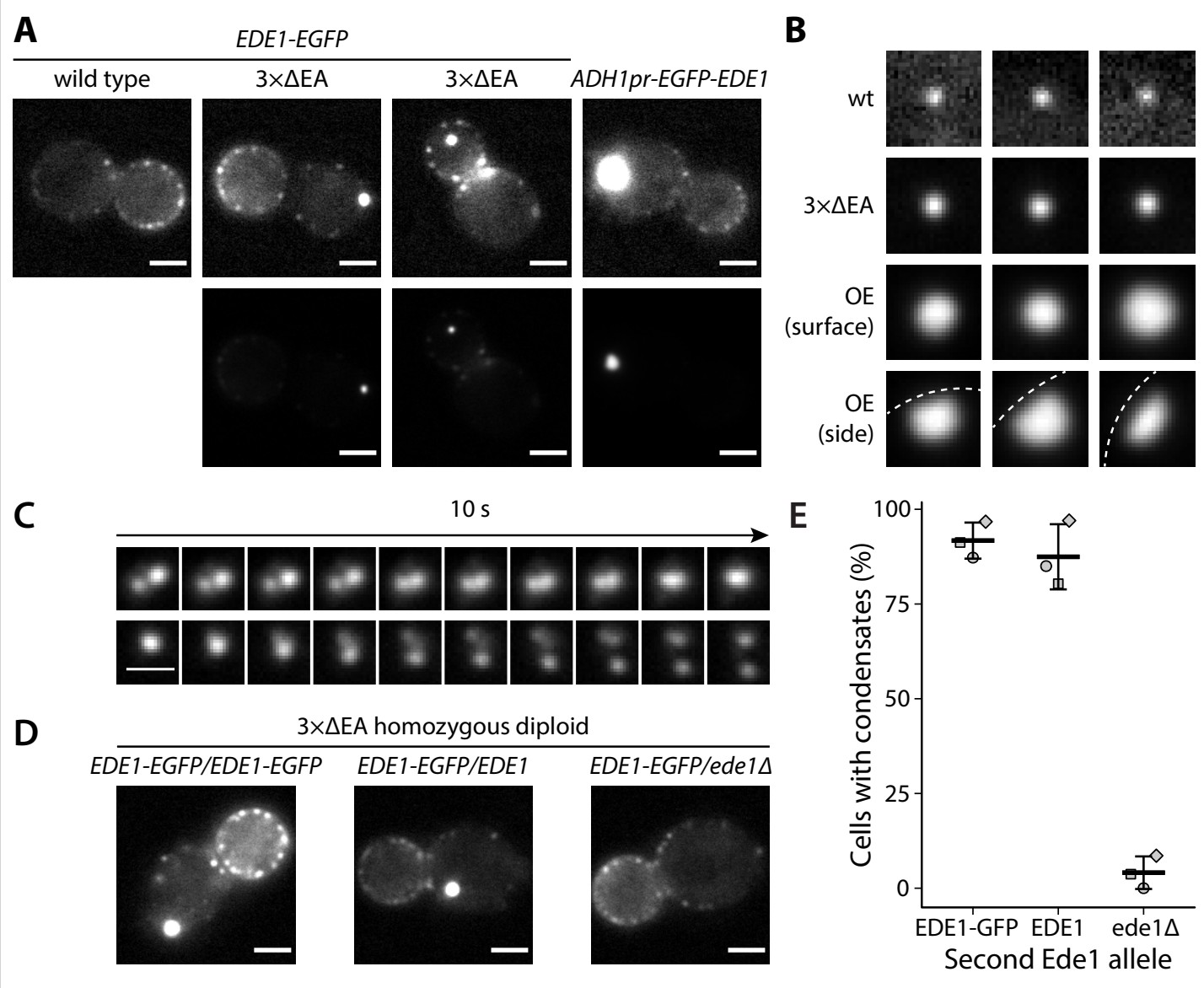

**Figure 1.** Excess cytosolic Ede1 assembles into condensates in vivo. (**A**) Representative images of yeast cells expressing Ede1-EGFP in wild-type (wt) and 3×ΔEA genetic backgrounds, or overexpressing EGFP-Ede1 under the control of the *ADH1* promoter. Mutant cell micrographs are shown using the same display range as the wt (top) or their full display range (bottom). Two cells are shown for 3×ΔEA background to display the membrane-associated and cytoplasmic localizations of Ede1 condensates. Scale bars: 2 µm. (**B**) Representative images of Ede1-EGFP at endocytic sites in wt background, Ede1-EGFP condensates in 3×ΔEA cells, and EGFP-Ede1 overexpression-induced condensates (OE). OE condensates are shown in two different orientations. Each frame is 1.5 µm × 1.5 µm; dotted white line represents the approximate position of the plasma membrane. (**C**) Two time series of Ede1-EGFP condensates undergoing apparent fusion (top) and fission (bottom) events. Scale bar: 1 µm. (**D**) Representative images of diploid cells homozygous for the 3×ΔEA background, each expressing Ede1-EGFP and differing in the second Ede1 locus: *EDE1-EGFP*, *EDE1*, or *ede1Δ*. Scale bars: 2 µm. (**E**) The fraction of cells containing condensates in each strain from panel D. Bars and whiskers show mean ± SD of three independent experiments. A range of 40–70 cells were analyzed per data point.

The online version of this article includes the following video, source data, and figure supplement(s) for figure 1:

**Source data 1.** Source data (panel E).

**Figure supplement 1.** Representative 3×ΔEA cells expressing Ede1 tagged with mNeonGreen, msGFP2, or mCherry.

**Figure 1—video 1.** A 1-hour movie of Ede1-EGFP in 3×ΔEA cells.

https://elifesciences.org/articles/72865/figures#fig1video1

the Ede1 overexpression strain were often large enough that their shape was resolvable (*Figure 1B*). They appeared circular in surface view, and as dome-like structures limited by the plasma membrane in the side view. The Ede1 condensates were remarkably long lived and we have observed individual condensates for up to 1 hour (*Figure 1—video 1*). This stands in contrast with normal endocytic sites, where Ede1-EGFP typically persists for 1-2 minutes (*Stimpson et al., 2009*). Despite their stability, some of the condensates appeared to undergo dynamic fission and fusion events, suggesting that they are not solid aggregates (*Figure 1C*).

All three adaptors absent from 3×ΔEA cells interact with Ede1, as well as membrane lipids and protein cargo. As Ede1 does not have known membrane-binding activity, it is likely that the cytosolic pool of Ede1 is increased both in the overexpression and the 3×ΔEA backgrounds, and the excess protein assembles into the condensates. To test whether the formation of condensates in 3×ΔEA background depends on Ede1 concentration, we generated diploid cells homozygous for the three adaptor deletions. We expressed Ede1-EGFP in these cells either from both *EDE1* alleles (*EDE1-EGFP/EDE1-EGFP*), or from one allele in combination with untagged *EDE1* (*EDE1-EGFP/EDE1*) or a deletion of *EDE1* (*EDE1-EGFP/ede1Δ*). The condensates formed in both strains expressing two alleles of *EDE1*, but not in the strain where only one *EDE1* allele was present (*Figure 1D*). This result suggests that the condensate assembly depends on Ede1 concentration.

The EGFP tag used in these experiments has a weak tendency to dimerize and can induce protein clustering in some in vivo contexts (*Costantini et al., 2012*). We expressed Ede1 tagged with monomeric fluorescent proteins mNeonGreen, msGFP2, and mCherry in 3×ΔEA cells and observed the same phenotype as with EGFP-tagged Ede1 (*Figure 1—figure supplement 1*). The observed phenotype is therefore unlikely to have been caused by the choice of the fluorescent tag.

It must also be noted that there is a large difference in brightness between genuine endocytic sites and condensates in mutant cells. Therefore, we chose to saturate the display of those images which show both classes of objects simultaneously, in order for the cells and endocytic sites to remain visible. This difference is demonstrated in panel A of *Figure 1*.

## Ede1 condensates exhibit liquid-like properties

Because of their spherical shapes, concentration dependence, and dynamic behaviors, we hypothesized that the Ede1 condensates might be phase-separated liquid droplets. To test this idea, we first performed fluorescence recovery after photobleaching (FRAP) experiments on Ede1-EGFP condensates in the 3×ΔEA background. After photobleaching, the condensates rapidly recovered most of their fluorescence (*Figure 2A*). The recovery half-time of a single-exponential FRAP model fitted to an average of 36 events was 22 s, and the mobile fraction was 63%.

We then examined the recovery of Ede1 in normal endocytic sites, photobleaching them during total internal reflection fluorescence (TIRF) imaging. We found that Ede1-EGFP also turns over fast at endocytic sites (half-time of 7.8 s and mobile fraction of 91%). The turnover at the endocytic sites was faster and the mobile fraction higher than in the condensates.

We also performed experiments where we bleached the Ede1 condensates only partially, in order to visualize diffusion of fluorescent molecules within the condensates. For these experiments we used cells overexpressing Ede1-EGFP, in which the condensates are larger than in the 3×ΔEA cells. When a subregion of an endocytic condensate was bleached, the fluorescence in the bleached and unbleached regions equalized within several seconds (*Figure 2C and D*; *Figure 2—video 1*).

The formation of phase-separated condensates depends on protein concentration and can be affected by environmental factors such as temperature (*Molliex et al., 2015*; *Nott et al., 2015*; *Franzmann et al., 2018*). When the 3×ΔEA cells cultured at 24°C were incubated at 37°C for 5 min, the condensates dissolved while the endocytic sites persisted (*Figure 2D*). When the temperature was raised to 42°C, Ede1 signal became entirely diffuse. This effect was reversible for both the endocytic sites, which reformed after several minutes, and the condensates, which reappeared within 30 min after return to 24°C.

1,6-Hexanediol is an aliphatic alcohol that disrupts weak protein-protein interactions (*Patel et al., 2007*) and is used to distinguish between solid and liquid protein aggregates (*Kroschwald et al., 2015*). Both Ede1 condensates and endocytic patches rapidly disappeared in 3×ΔEA cells upon 1,6-hexanediol treatment (*Figure 2E*). Interestingly, endocytic patches only fully dissolved at higher 1,6-hexanediol concentrations than Ede1 condensates.

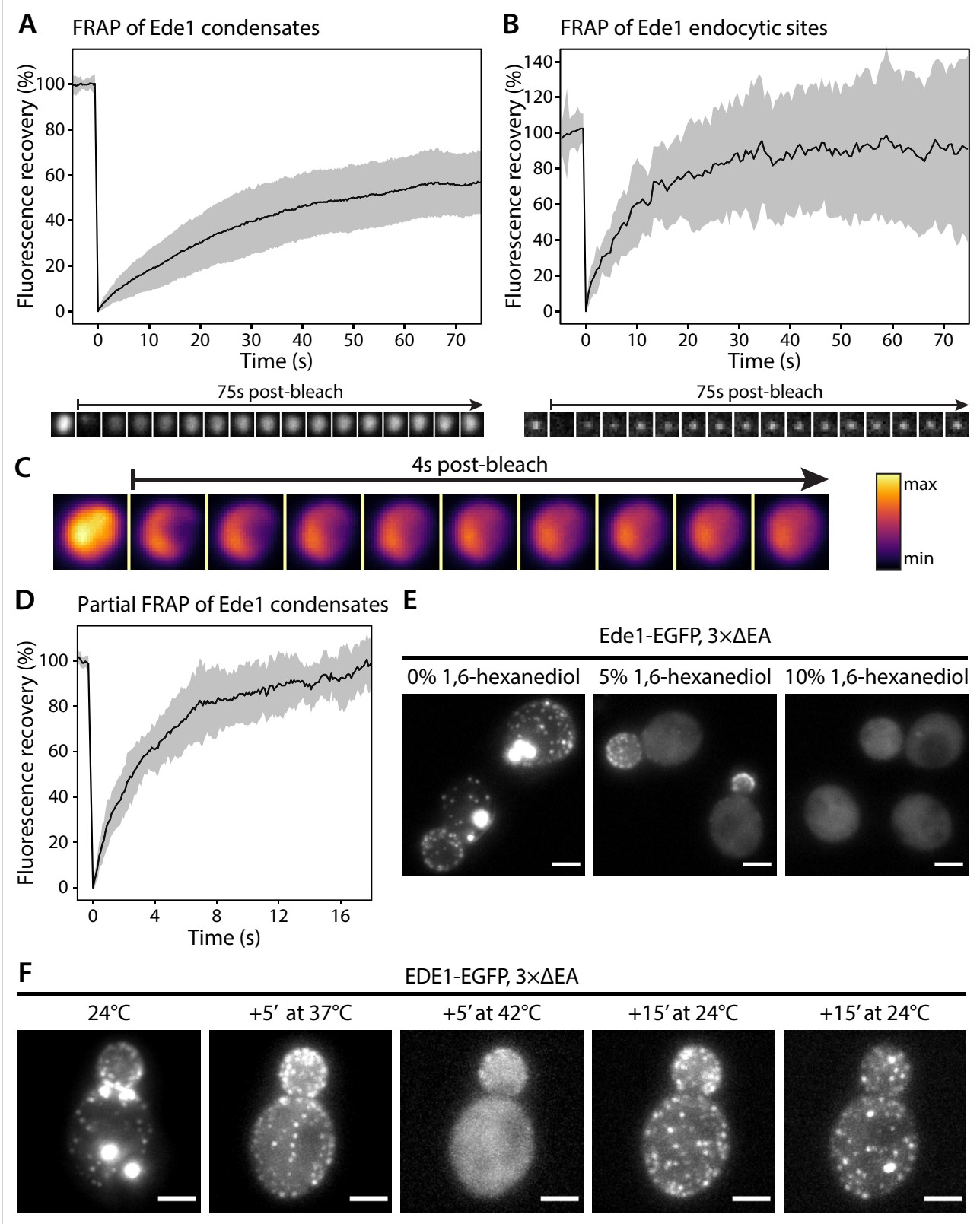

**Figure 2.** Ede1 structures exchange molecules with the cytoplasm and respond to temperature changes. (**A, B**) Fluorescence recovery after photobleaching (FRAP) of Ede1-EGFP condensates in 3×ΔEA cells (**A**) and endocytic sites in normal cells (**B**). Plots show mean fluorescence recovery ± SD; n = 36 across four independent experiments (panel A) and n = 14 across three independent experiments (panel B). Representative time series are shown below each plot. Each frame is 1 μm × 1 μm. (**C**) Time series of a partial bleaching of a condensate in a cell overexpressing EGFP-Ede1. A

*Figure 2 continued on next page*

*Figure 2 continued*

perceptually uniform color lookup table has been applied to highlight the changes in intensity. Each frame is 1.5 µm × 1.5 µm.(**D**) Average fluorescence recovery (n = 14) after partial bleaching of condensates in cells overexpressing EGFP-Ede1, as in panel (**C**). (**E**) Representative cells after 5-min treatment with indicated concentrations of 1,6-hexanediol. Maximum Z-projections. (**F**) Ede1-EGFP was imaged in 3×ΔEA cells at different temperatures. Cells were grown and imaged at 24°C. The temperature was raised to 37 and 42°C and returned to 24°C for the indicated amounts of time. Maximum Z-projections. All scale bars: 2 µm.

The online version of this article includes the following video and source data for figure 2:

**Source data 1.** Fluorescence recovery data (panel A).

**Source data 2.** Fluorescence recovery data (panel B).

**Source data 3.** Fluorescence recovery data (panel D).

**Figure 2—video 1.** Partial condensate bleaching in Ede1 overexpression cells.

https://elifesciences.org/articles/72865/figures#fig2video1

Taken together our results show that Ede1 both in the condensates and at the endocytic sites behaves in a highly dynamic, liquid-like manner.

Next, we sought to determine the concentration of Ede1 in the 3×ΔEA and Ede1 overexpression cells relative to the wild-type cells. We used spinning-disk confocal microscopy to limit the influence of out-of-focus condensates on the quantification of cytosolic intensity. We quantified the mean pixel intensity in entire cell volumes and in small cytosolic regions of cell cross-sections (*Figure 3A*). The total cellular intensity was on average 122 and 321% of the wild-type in 3×ΔEA and overexpression cells, respectively. The cytosolic intensity was nevertheless uniform across all three strains and we could detect no statistically significant differences (p=0.45 in F-test).

We also induced overexpression of EGFP-tagged Ede1 using the weakened galactokinase promoter GALS (*Mumberg et al., 1994*). We followed the changes in fluorescence intensity over 8 hr after switching carbon source from glucose to galactose (*Figure 3B and C*; *Figure 3—video 1*). The cells, in which overexpression was induced after glucose repression, initially showed no Ede1 sites at the membrane. After several hours, Ede1 appeared as transient endocytic sites, and later formed large and stable condensates. The intensity of the cytosolic regions in the overexpression cells never surpassed the cytosolic intensity of wild-type cells, even as the total intensity of the mutant reached approximately 150% of wild-type intensity by the end of the experiment.

These experiments suggest that the cytosolic concentration of Ede1 is buffered by the formation of the condensates. Moreover, the cytoplasmic fluorescence intensity in wild-type cells is already at the limit observed during overexpression. This signifies that the total concentration in wild-type cells is above the critical concentration required for phase separation.

## Ede1 condensates recruit other endocytic proteins

We then imaged double-tagged strains to test whether other endocytic proteins colocalize with Ede1 condensates in the 3×ΔEA background (*Figure 4A and B*). We found that the condensates contain multiple early (Syp1, Ent1, Sla2) and late (End3, Pan1, Sla1) coat proteins, as well as the actin nucleation-promoting factor Las17, known to physically interact with the End3/Pan1/Sla1 complex (*Sun et al., 2015*; *Feliciano and Di Pietro, 2012*).

Three proteins—Myo5, Abp1, and Rvs167—whose arrival overlaps with actin polymerization at the endocytic sites (*Sun et al., 2006*) localized to a minority of the condensates (42, 20, and 10%, respectively). We further examined the interaction of condensates and Abp1 using timelapse imaging and found that Abp1-mCherry patches assembled on the condensates transiently (*Figure 4C*), which explains the partial colocalization. Abp1-mCherry was also present whenever the apparent fission of condensates occurred (*Figure 4C*, *Figure 4—video 1*). The appearance of Abp1 during condensate fission suggests that fission could be caused by a force exerted by actin filaments. Some of the proteins we see in the condensates, such as Las17, could potentially trigger actin polymerization.

Costaining with FM4-64 showed that endocytic protein condensates do not contain a significant membrane fraction (*Figure 4—figure supplement 1*).

Overall, the colocalization experiments showed that the endocytic condensates are complex, and that the proteins contained within them are at least partially functional as they can recruit their interaction partners and are associated with cycles of assembly and disassembly of actin.

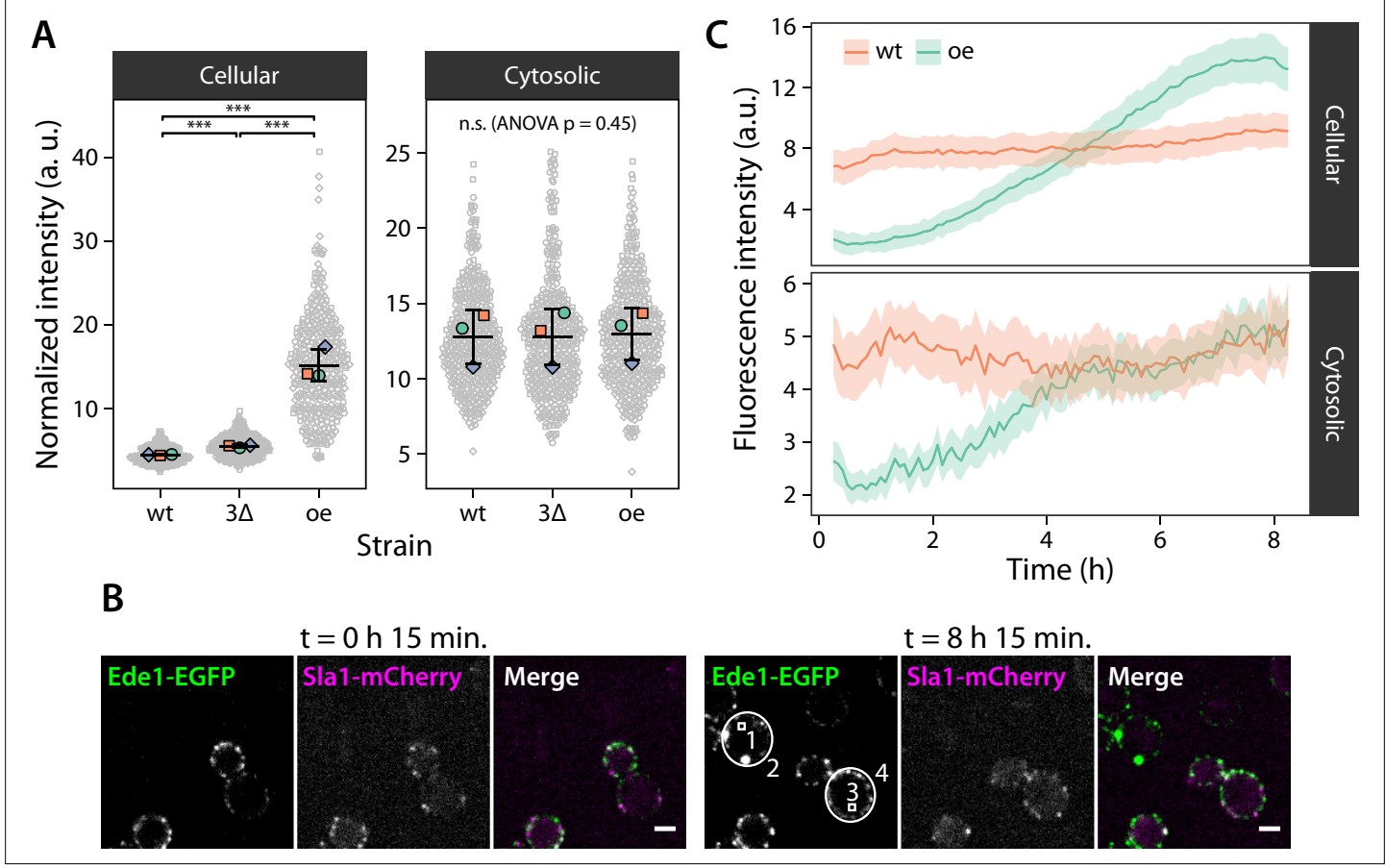

**Figure 3.** Condensate formation limits the concentration of Ede1 in the cytoplasm. (**A**) Fluorescence intensity was measured in wild-type (wt) and 3×ΔEA cells expressing Ede1-EGFP, or cells overexpressing (oe) EGFP-Ede1 from the ADH1 promoter. Gray points represent mean pixel intensities of entire cell volumes (*Cellular*), or small regions manually selected from cell cross-sections (*Cytosolic*). Large points: mean values from independent replicates; central line and whiskers: mean ± SD of replicate means. Pairwise comparisons based on a linear mixed model (n.s., not significant; ***, p<0.001). (**B**) Two yeast strains were imaged for 8 hr after change of carbon source from glucose to galactose (see Materials and methods). Cells express EGFP-Ede1 under the control of GALS promoter (green channel only), or Ede1-EGFP and Sla1-mCherry expressed from the endogenous loci (green and magenta, respectively). Scale bars: 2 μm. (**C**) Fluorescence intensity during the expression induction was measured in regions representing entire cells (2 and 4 in panel B) and their cytoplasm (1 and 3). Mean intensity is shown for endogenously expressed (wt) or overexpressed (oe) Ede1 after background subtraction, ±2 × SEM (n = 40 cells for each strain).

The online version of this article includes the following video and source data for figure 3:

**Figure 3—video 1.** Movie of cells shown in panel B.

https://elifesciences.org/articles/72865/figures#fig3video1

**Source data 1.** Source data, code, and statistical details (panel A).

**Source data 2.** Source data and code (panel C).

## Ede1 central region is necessary and sufficient for phase separation

Ede1 is a 1381 amino acid long, multidomain protein (*Figure 5A*). Its N-terminal region contains three Eps15-homology (EH) domains that interact with asparagine-proline-phenylalanine motifs found on endocytic adaptors such as Ent1/2, Yap1801/2, and Sla2 (*Maldonado-Báez et al., 2008*). Such repeats of domains interacting with linear motifs are known to promote liquid-liquid phase separation (*Li and Elledge, 2012*; *Banjade and Rosen, 2014*). The EH domains are followed by a proline-rich region and a coiled-coil domain (*Reider et al., 2009*; *Lu and Drubin, 2017*). The C-terminal half of Ede1 contains a Syp1-interacting region (*Reider et al., 2009*) and a ubiquitin-associated (UBA) domain.

We noticed that the proline-rich region contains a high number of asparagine and glutamine residues, a hallmark of prion-like domains (PLDs) which are proposed to regulate phase separation of many proteins (*Alberti et al., 2009*; *Franzmann et al., 2018*; *Franzmann and Alberti, 2019*). We

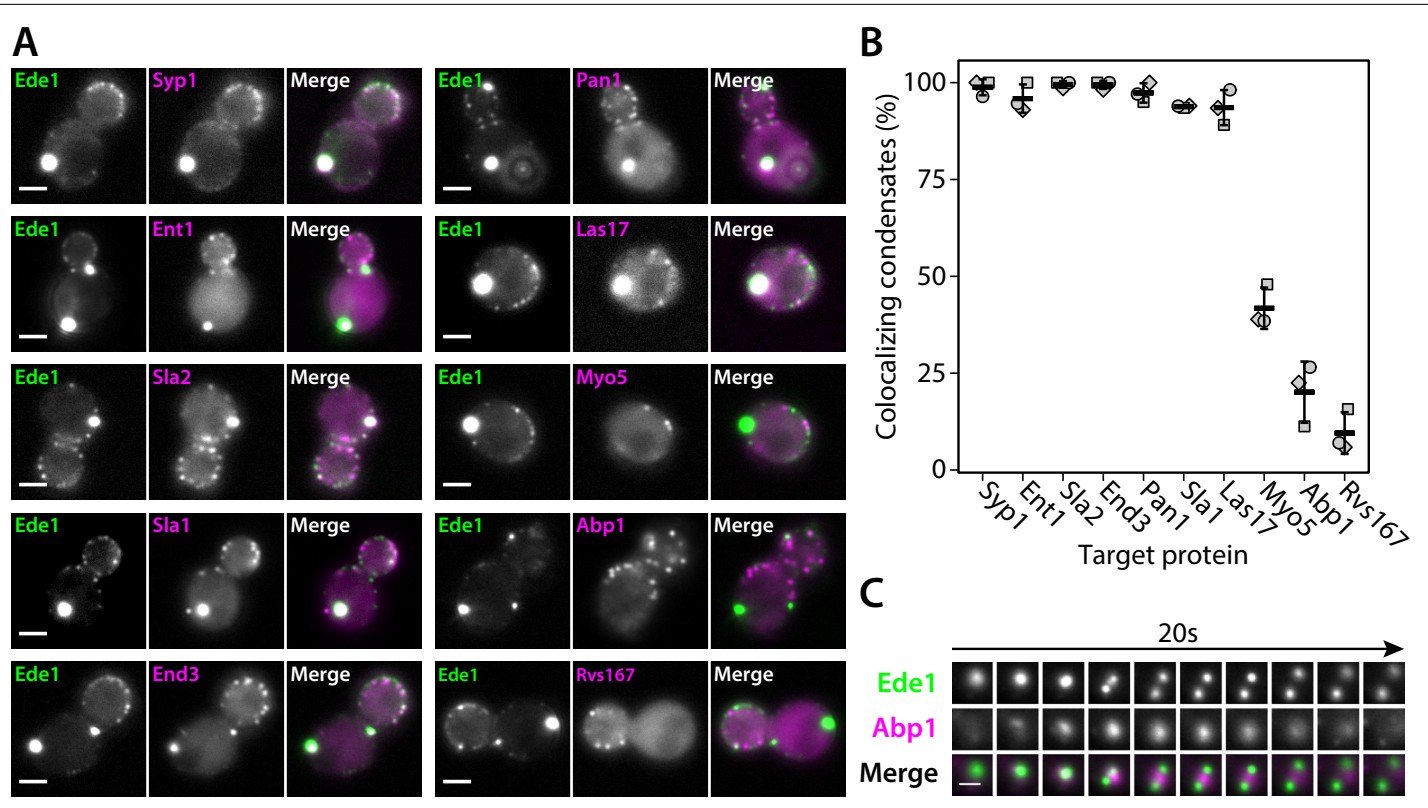

**Figure 4.** Endocytic condensates recruit many proteins. (**A**) Images of representative cells expressing Ede1-EGFP and indicated endocytic proteins tagged with mCherry in 3×ΔEA background. (**B**) Fraction of Ede1-EGFP condensates that colocalized with mCherry puncta in each strain from panel A. Bars and whiskers show mean ± SD of three independent experiments. A range of 36–98 cells were analyzed per data point. (**C**) Montage from timelapse imaging of Ede1-EGFP and Abp1-mCherry during apparent fission of an Ede1 condensate.

The online version of this article includes the following video, source data, and figure supplement(s) for figure 4:

**Source data 1.** Source data (panel B).

**Figure supplement 1.** Ede1 condensates do not colocalize with membranes stained by FM4-64.

**Figure 4—video 1.** A 2-min movie of Ede1-EGFP (green) and Abp1-mCherry (magenta) in 3×ΔEA background showing repeated transient localization of Abp1 to Ede1 condensates, and an example of Abp1 recruitment coinciding with apparent condensate fission and subsequent fusion.
https://elifesciences.org/articles/72865/figures#fig4video1

used PLAAC (*Lancaster et al., 2014*), a web-based version of the algorithm used by *Alberti et al., 2009* to detect prion candidates in yeast proteome, to analyze the Ede1 sequence (*Figure 5A*). The algorithm detected a 99-amino acid-long prion-like sequence between amino acids 374 and 472, suggesting that this region could also be involved in the phase separation of endocytic proteins. We also consulted IUPred2a (*Mészáros et al., 2018*) and MobiDB-lite (*Necci et al., 2017*) algorithms to predict intrinsically disordered regions (IDRs) in Ede1. About 36% of Ede1 is predicted to be disordered; the unstructured regions are contained within the proline- and glutamine-rich region, and between the coiled-coil and the UBA domain.

To understand which features of Ede1 mediate phase-separation, we expressed a series of truncations of Ede1 in both wild-type and 3×ΔEA backgrounds, and analyzed their localization to condensates and endocytic sites (*Figure 5B and C*). In our 3×ΔEA background, the N- and C-terminal regions were dispensable for condensate formation. Surprisingly, the Ede1 central region consisting of amino acids 366–900 (the unstructured PQ-rich region and the coiled-coil domain) localized to large condensates in both the 3×ΔEA and wild-type backgrounds. It was also the minimal construct to form these condensates, as constructs containing only the coiled-coil or the PQ-rich region showed diffuse cytoplasmic localization.

For the wild-type background, our results are largely consistent with previously published results about the localization of Ede1 mutants (*Boeke et al., 2014*; *Lu and Drubin, 2017*). Namely, the

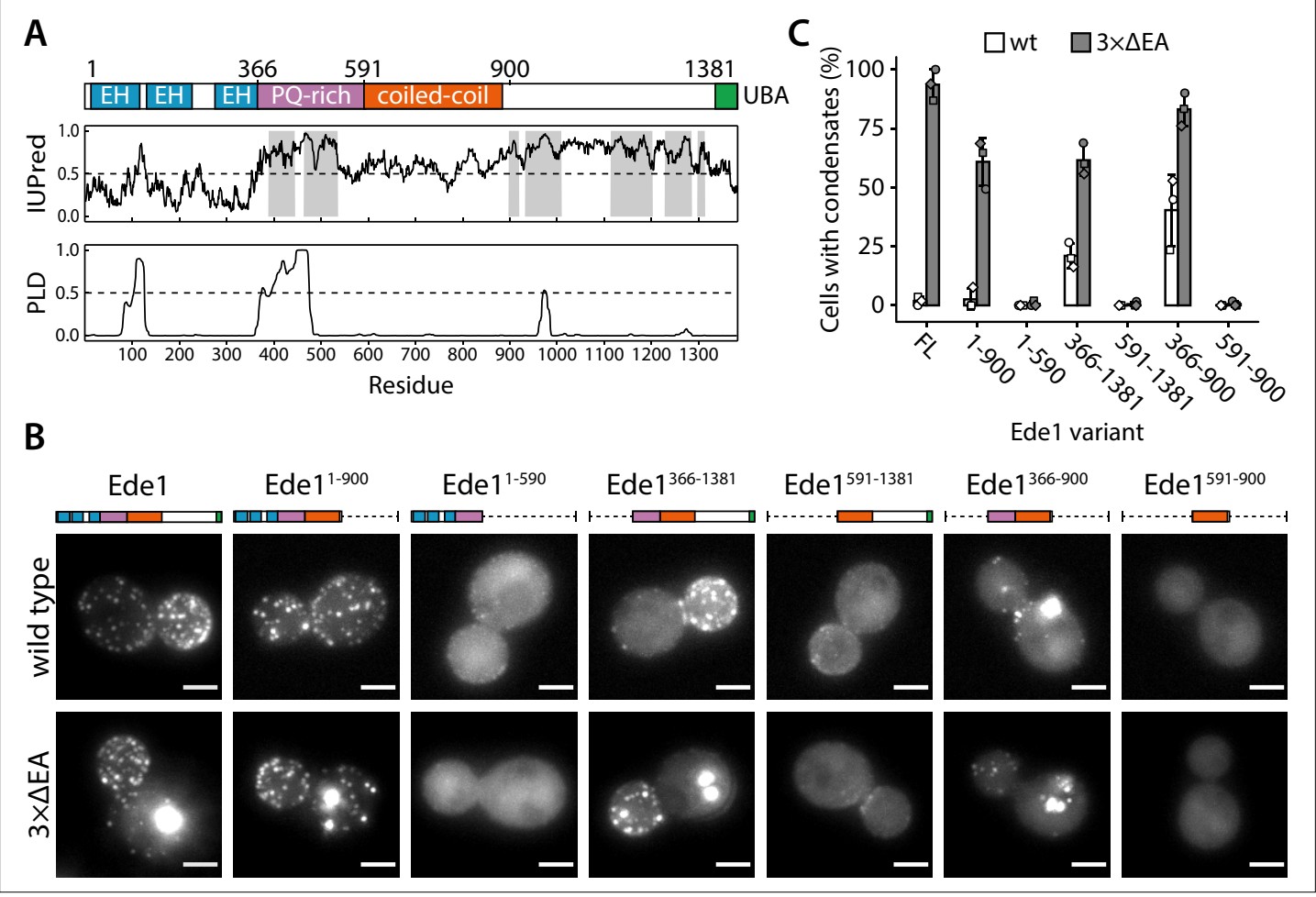

**Figure 5.** The central region of Ede1 is necessary and sufficient for condensate formation. (**A**) The domain structure of Ede1 and prediction of disordered and prion-like regions. EH, Eps15-homology domain; UBA, ubiquitin-associated domain. Domains are drawn to scale according to UniProt entry P34216, and numbers above mark domain boundaries used in our constructs. The top plot represents IUPred2a disorder prediction score, with the shaded areas predicted to be disordered by MobiDB-lite consensus method. Prion-like domain (PLD) prediction score was calculated using the PLAAC software. (**B**) Representative cells expressing full-length (FL) Ede1 and its truncation mutants in wild-type and 3×ΔEA backgrounds. All constructs are C-terminally tagged with EGFP. Maximum intensity projections of 3D volumes, scale bars: 2 μm. (**C**) The fraction of cells containing condensates in each strain from panel B. Bars and whiskers show mean ± SD of three independent experiments. A range of 37–78 cells were analyzed per data point.

The online version of this article includes the following source data and figure supplement(s) for figure 5:

**Source data 1.** Source data (panel C).

**Figure supplement 1.** Levels of Ede1 mutants assessed by western blotting.

**Figure supplement 1—source data 1.** Source data (western blotting).

coiled-coil domain was necessary for Ede1 to assemble into endocytic sites, while the N- and C-terminal parts of Ede1 were individually dispensable.

We analyzed the concentration of truncated variants in both backgrounds by quantitative western blotting (*Figure 5—figure supplement 1*). The truncated Ede1 variants were expressed at higher levels than the full-length protein. This observation suggests that the loss of condensates in strains with truncated Ede1 is indeed caused by missing motifs rather than lowered concentration. We also confirmed that the concentration of full-length Ede1 increased in the 3×ΔEA background. It is unclear whether this represents a compensatory genetic mechanism, or if condensation is initially caused by reduced plasma membrane recruitment and subsequently interferes with protein degradation.

## The functional significance of the Ede1 central region

To test the role of the central region of Ede1 in endocytosis, we created EGFP-tagged Ede1 mutants with internal deletions of amino acids 366–590 (Ede1$^{\Delta PQ}$), 591–900 (Ede1$^{\Delta CC}$), and 366–900 (Ede1$^{\Delta PQCC}$). Ede1$^{\Delta PQCC}$ failed to localize to endocytic sites, whereas the two single-domain Ede1$^{\Delta PQ}$ and Ede1$^{\Delta CC}$ deletion mutants were still punctate, but more diffuse compared to full-length Ede1-EGFP (*Figure 6A*).

Next, we tested if these Ede1 mutants had endocytic defects by using Sla1 as a reporter of the late phase of endocytosis. We tagged Sla1 with EGFP in Ede1 mutant strains (*Figure 6B*) and quantified the density and lifetimes of endocytic sites (*Figure 6C and D*). In *ede1Δ* and *ede1$^{\Delta PQCC}$* cells, the mean number of endocytic events marked by Sla1-EGFP per µm² was reduced by 46 and 43% of the wild-type, respectively. Consistent with their effects on Ede1 recruitment, the *ede1$^{\Delta PQ}$* and *ede1$^{\Delta CC}$* mutations caused intermediate reduction in patch density (by 24 and 26%, respectively). All differences from the wild type were statistically significant (p<0.001 in Tukey-Kramer test). The difference between *ede1Δ* and *ede1$^{\Delta PQCC}$* was not statistically significant (p=0.86).

The Sla1 lifetimes were likewise affected by the deletion of Ede1 central region. In *ede1Δ*, Sla1-EGFP lifetime was decreased by 29% and in *ede1$^{\Delta PQCC}$*, by 28%. The deletions of individual regions again showed intermediate defects (13 and 18% for Ede1$^{\Delta PQ}$ and Ede1$^{\Delta CC}$, respectively). All differences from the wild type were statistically significant (p<0.01 in Tukey-Kramer test). The deletion of the entire *EDE1* gene was not significantly different from the *ede1$^{\Delta PQCC}$* mutant (p = 0.99).

The Ede1$^{\Delta PQCC}$ mutant does not localize to endocytic sites. We wanted to test if the N- and C-terminal domains alone could support endocytosis if a strong interaction with another endocytic protein was introduced. We generated cells coexpressing Ede1$^{\Delta PQCC}$-mCherry-FKBP and Syp1-FRB or Sla2-FRB in order to link the Ede1 mutant to another component of the early coat via the rapamycin-inducible FKBP-FRB dimerization system (*Haruki et al., 2008*). Recruitment to Syp1 caused the Ede1$^{\Delta PQCC}$-mCherry-FKBP signal to become more prominent around the bud necks, but did not rescue membrane patch formation, whereas recruitment to Sla2 partially rescued the patch localization of Ede1$^{\Delta PQCC}$ (*Figure 6—figure supplement 1*). The average Sla1 density did not significantly change in either of these yeast strains upon rapamycin treatment (p=0.53 and p=0.59 between treated and untreated cells in Syp1-FRB and Sla2-FRB strains, respectively).

Taken together, our results show that the central region is essential for Ede1 to promote efficient endocytosis and to regulate the timing of coat maturation.

In *ede1Δ* cells, many of the early endocytic proteins fail to localize to endocytic sites (*Stimpson et al., 2009*; *Carroll et al., 2012*). We therefore visualized the localization of early proteins in Ede1 mutants lacking the central region.

Different proteins were affected by the Ede1 central deletions in different ways (*Figure 7*), consistent with the work done on *ede1Δ* mutants. The localization of Apl1 (β-subunit of the AP-2 complex) was the most severely disrupted. Apl1-EGFP patches were less defined in Ede1$^{\Delta PQ}$ background, and undetectable in Ede1$^{\Delta CC}$ or Ede1$^{\Delta PQCC}$ cells. In these cells, Apl1-EGFP signal was dispersed in the cytoplasm, with a faint presence around the bud neck. Syp1 and Yap1801 remained localized to the membrane in all of the mutants, but the signal became more diffuse along the membrane and the patches less defined. This effect was the strongest for Ede1$^{\Delta PQCC}$, with intermediate effects in Ede1$^{\Delta PQ}$ and Ede1$^{\Delta CC}$ cells. However, Ent1 and Sla2 still assembled into endocytic patches in all of the Ede1 mutants. Taken together, our results indicate that the Ede1 central region is essential to concentrate early endocytic proteins.

## Ede1 central region can be replaced by other prion-like domains

We wanted to test whether the loss of function in *ede1$^{\Delta PQCC}$* cells in respect to Sla1 density could be rescued by heterologous protein sequences, such as IDRs, globular domains, or coiled-coils (*Figure 8A*).

First, we replaced amino acids 366–900 of Ede1 with different IDRs. We considered three factors for choosing the replacement IDRs: known phase separation activity, Q/N content, and similarity to the Ede1 IDR sequence. We chose the region of Sup35 spanning the N and M regions based on its known phase separation activity in yeast cells (*Franzmann et al., 2018*). We chose the IDR of Snf5 because of its high score in the prion-like screen by *Alberti et al., 2009*, reflecting a Gln-rich amino acid sequence, and the IDR of Whi3, because the method developed by *Zarin et al., 2019* indicated it as one of the sequences most similar to the PQ-rich region of Ede1. Finally, we also included the

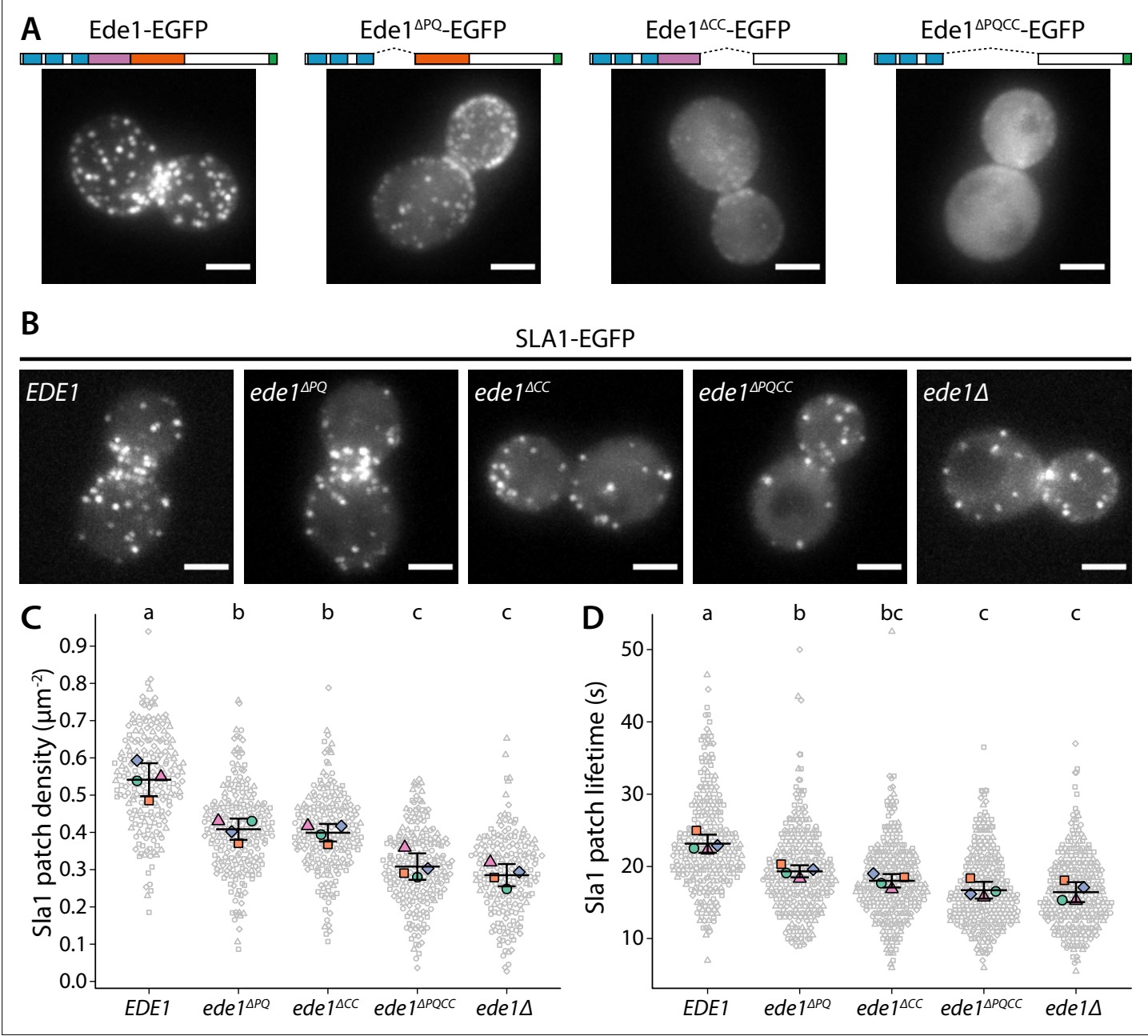

**Figure 6.** Ede1 features necessary for phase separation are also crucial for its function. (**A**) Representative cells expressing full-length Ede1 and three internal Ede1 deletion mutants: Ede1$^{ΔPQ}$ (Δ366-590), Ede1$^{ΔCC}$ (Δ591-900), and Ede1$^{ΔPQCC}$ (Δ366-900) tagged with EGFP. (**B**) Representative cells expressing Sla1-EGFP and indicated Ede1 mutants. (**C, D**) Sla1 patch density and lifetime in Ede1 mutants. Large points represent mean measurements from independently repeated datasets. Central line and whiskers denote the mean ± SD calculated from dataset averages. Gray points show individual observations. Letters denote pairwise comparisons based on Tukey-Kramer test; groups which do not share any letters are significantly different at α = 0.05. Scale bars: 2 μm.

The online version of this article includes the following source data and figure supplement(s) for figure 6:

**Source data 1.** Source data, code, and statistical details (panel C).

**Source data 2.** Source data, code, and statistical details (panel D).

**Figure supplement 1.** Recruitment of Ede1$^{ΔPQCC}$ to other proteins cannot rescue the endocytic defect.

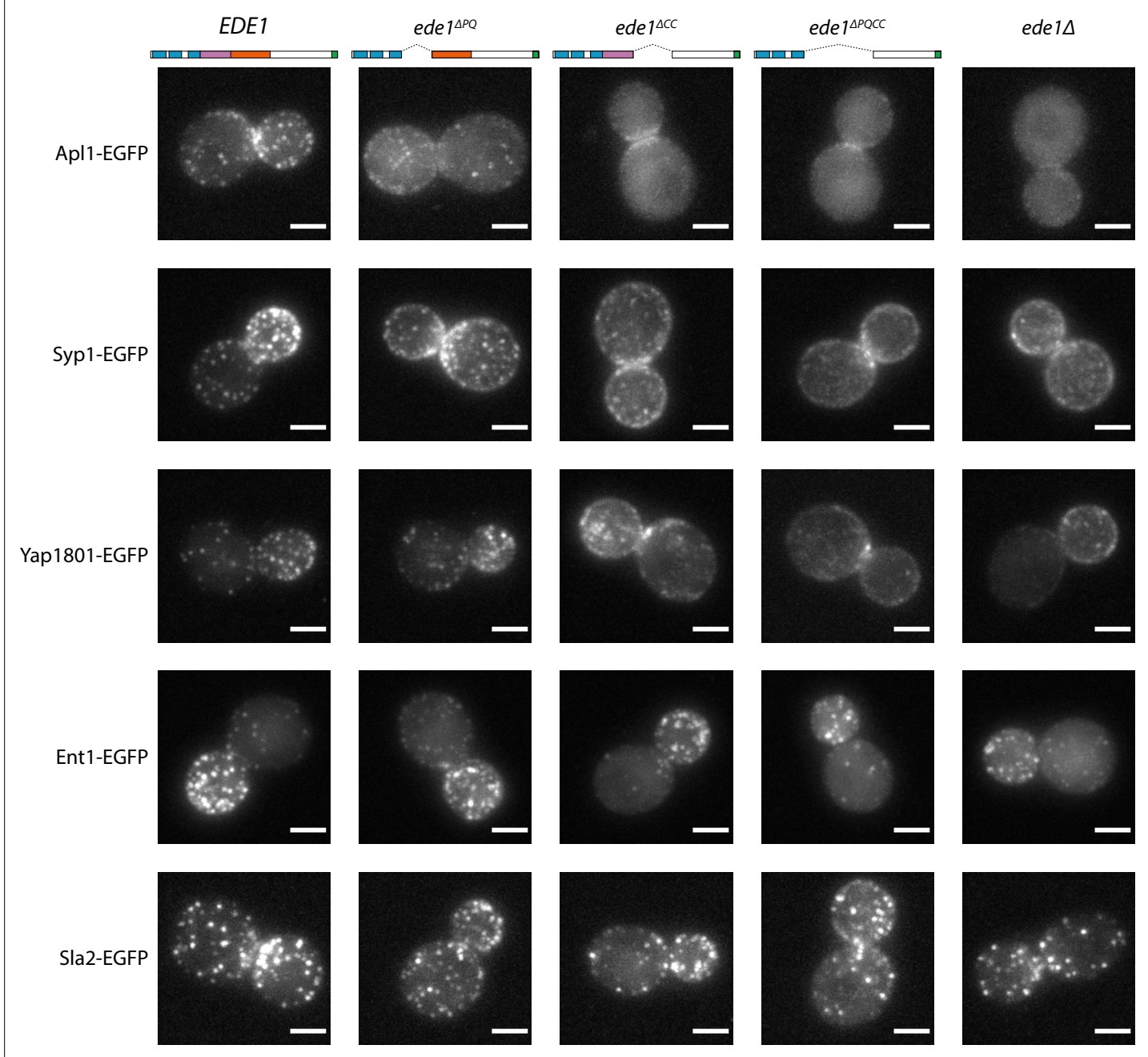

**Figure 7.** Ede1 central deletion mutants are defective in early protein localization. Maximum-intensity projections of 3D volumes are shown for representative cells with different early proteins tagged with EGFP. The strains express Ede1 species indicated at the top. Scale bars: 2 μm.

low-complexity domain of human FUS for its well-known tendency to phase separate and form hydrogels (*Patel et al., 2015*; *Kato et al., 2012*).

We also replaced the central region of Ede1 with several structured domains. We used the red fluorophores mCherry and dTomato as globular linkers assuming respectively monomeric and dimeric states. We also used the coiled-coil domains from two kinesin motors as oligomeric rod-shaped linkers: the kinesin-1 heavy chain (Khc amino acids 335–931) from *Drosophila melanogaster* and the human kinesin-5 Eg5 (amino acids 358–797), which form dimers and tetramers in their respective contexts (*de Cuevas et al., 1992*; *Scholey et al., 2014*).

All of the Ede1 central region replacement constructs were expressed from the endogenous genomic locus under the control of the native promoter.

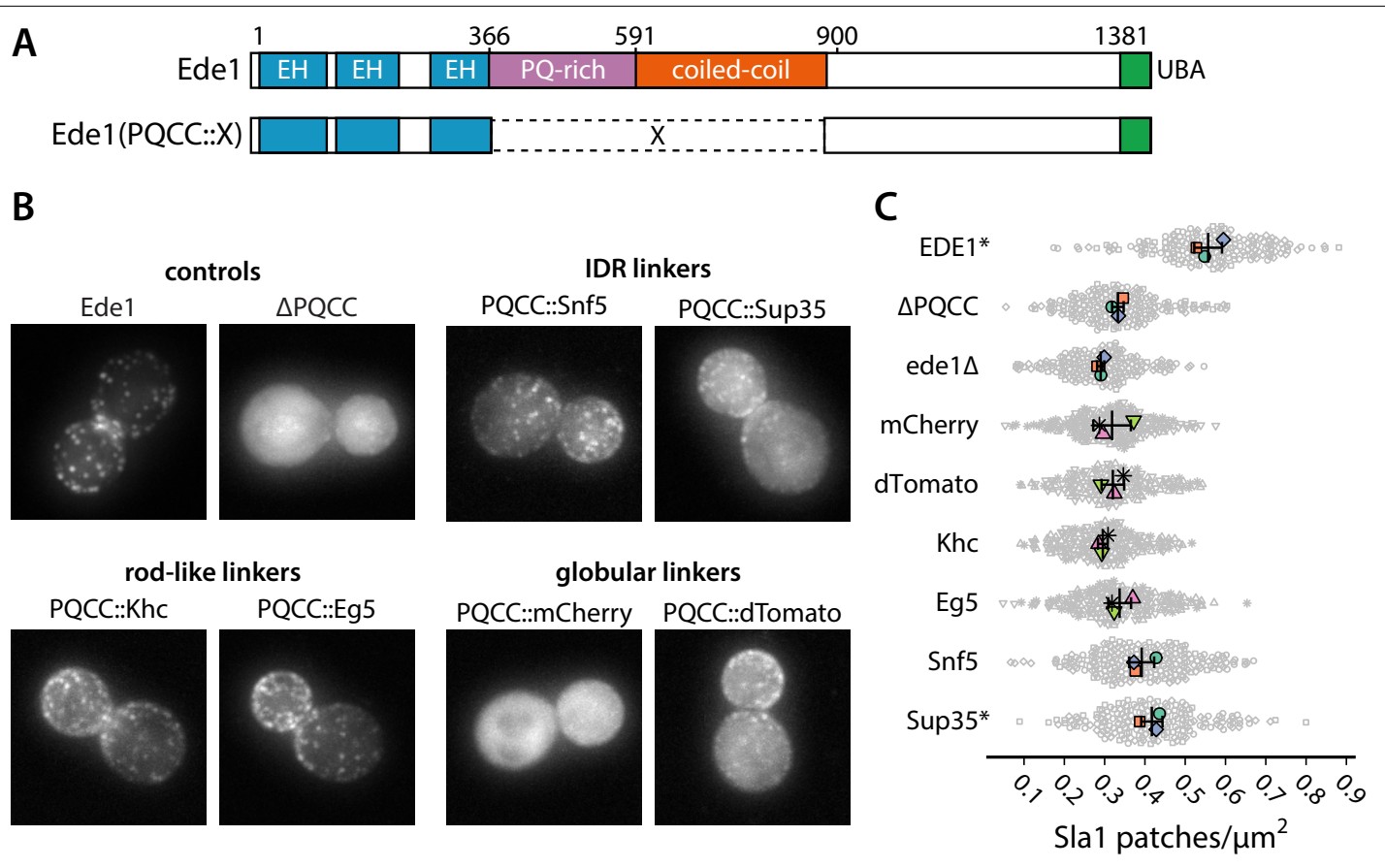

**Figure 8.** Prion-like domains can partially replace Ede1 central region. (**A**) Domain structure of the Ede1 central region replacement constructs. Amino acids 366–900 of Ede1 were replaced with prion-like intrinsically disordered region (IDR) sequences, monomeric (mCherry) or dimeric (dTomato) fluorescent proteins, and dimeric (Khc) or tetrameric (Eg5) coiled-coils. All mutants were expressed from the Ede1 locus under the control of the native promoter. (**B**) Representative cells expressing indicated Ede1 mutants tagged C-terminally with msGFP2. (**C**) Quantification of Sla1-EGFP patch density in strains expressing indicated Ede1 mutants. Large points represent mean measurements from independently repeated datasets. Central line and whiskers denote the mean ± SD calculated from dataset averages. Gray points show individual cells from all datasets. At α = 0.05, all mutants are significantly different from wild type, and groups marked with an asterisk (*) are significantly different from Ede1^ΔPQCC (Tukey-Kramer test; a complete table of pairwise comparisons and effect sizes can be found in *Figure 8—source data 1*). All scale bars: 2 μm.

The online version of this article includes the following source data and figure supplement(s) for figure 8:

**Source data 1.** Source data, code, and statistical details (panel C).

**Figure supplement 1.** Ede1^FUS and Ede1^Whi3 intrinsically disordered region replacement constructs.

We assessed the localization of Ede1 constructs tagged with msGFP2 (*Figure 8B*, *Figure 8—figure supplement 1*). We found that the localization was partially rescued by the insertion of Whi3, Snf5, and Sup35 IDRs, as well as the oligomeric linkers (dTomato, Khc, Eg5), in place of the deleted central region. The localization was not rescued by the FUS low-complexity region or the monomeric globular linker mCherry.

We also assessed the density of Sla1-EGFP patches in cells expressing untagged Ede1^mCherry, Ede1^dTomato, Ede1^Khc, Ede1^Eg5, Ede1^Snf5, or Ede1^Sup35 (*Figure 8C*). All differences from wild type were statistically significant (p<0.001). No single replacement mutant was able to fully rescue the Sla1 patch density defect present in *ede1^ΔPQCC* cells. Only *ede1^Sup35* was able to significantly rescue the Sla1 density in respect to the deletion of Ede1 central region. In *ede1^ΔPQCC* and *ede1Δ* cells, the mean density of Sla1 patches was reduced by 40 and 48%, respectively. In *ede1^Sup35*, the mean density was only reduced by 25% from wild type, respectively.

These results support the hypothesis that prion-like domains can aid clustering of the endocytic proteins.

## Ede1 central region can cluster a heterologous lipid-binding protein

We hypothesized that the phase separation mediated by Ede1 central region is able to cluster membrane-associated proteins. To test our hypothesis, we fused the Ede1 central region to a diffusely membrane-bound protein. We created a GFP-Ede1$^{366-900}$-2×PH(PLCδ) construct, based on a phosphatidylinositol 4,5-bisphosphate (PI(4,5)P$_2$) probe developed by *Stefan et al., 2002*, by inserting the Ede1 central region between GFP and a tandem repeat of pleckstrin homology (PH) domain of phospholipase C δ1 (PLCδ). The original GFP-2×PH(PLCδ) construct is distributed homogeneously on the plasma membrane, while GFP-Ede1$^{366-900}$ alone localized to bright intracellular condensates. In contrast, the fusion construct localized to the plasma membrane, forming puncta that resembled endocytic sites (*Figure 9A,B*).

We also noticed subpopulations of cells with different localization patterns of the construct (*Figure 9B*). We speculated that the variable patterns were caused by heterogeneity in protein expression level due to plasmid copy number variation. To test that hypothesis, we expressed GFP-Ede1$^{366-900}$-2×PH(PLCδ) from centromeric plasmids containing four different promoters of increasing strength (*Mumberg et al., 1995*). We classified the localization of the construct in these cells as 'diffuse', 'punctate', or 'networked'. We found that the tendency to cluster into different patterns correlated with promoter strength and the expression level. Low expressing cells had more diffuse localization of the construct, and separated into puncta or well-separated regions as the concentration increased (*Figure 9C,D*).

The puncta formed by the GFP-Ede1$^{366-900}$-2×PH(PLCδ) construct were stable over long imaging periods, but dynamically recruited the late coat protein Sla1 (*Figure 9F*, *Figure 9—video 1*, *Figure 9—video 2*). Sla1-mCherry persisted at these sites with similar lifetimes as during normal endocytosis indicating that the condensates can recruit endocytic coat components. Unlike full-length Ede1 at the endocytic sites, the chimeric construct does not undergo cycles of assembly and disassembly. When we photobleached the structures formed by highly expressed GFP-Ede1$^{366-900}$-2×PH(PLCδ) we saw no recovery (*Figure 9—figure supplement 1A*), indicating that without the terminal domains, Ede1 central region might form solid, rather than liquid-like, structures. 10% 1,6-hexanediol also failed to dissolve the GFP-Ede1$^{366-900}$-2×PH(PLCδ) structures (*Figure 9—figure supplement 1B*). Structures formed by strongly overexpressed GFP-Ede1$^{366-900}$ were partially dissolved by 10% 1,6-hexanediol. This suggests that the stability of GFP-Ede1$^{366-900}$-2×PH(PLCδ) might also be modulated by the PH domains interacting with the membrane. In addition, we noticed that 1,6-hexanediol treatment caused membrane deformations, or possible clustering of the PH domains expressed without the Ede1 fusion (*Figure 9—figure supplement 1B*). This observation is consistent with the previous reports of wide-ranging effects of 1,6-hexanediol (*Kroschwald et al., 2015*), and underscores that 1,6-hexanediol experiments need to be interpreted with caution.

These results show that directing the Ede1 central region to the plasma membrane is sufficient to create puncta on the membrane in a concentration-dependent manner. Surprisingly, these long-lived sites can repeatedly recruit endocytic coat components.

## Discussion

Ede1 is the key organizer of the early phase of endocytosis in yeast (*Stimpson et al., 2009*; *Boeke et al., 2014*; *Lu and Drubin, 2017*). Our results indicate that the large clusters of Ede1 observed previously in mutant cells (*Boeke et al., 2014*) are in fact phase-separated protein droplets. Moreover, we found that the cytosolic concentration of Ede1 in normal cells is at the same critical limit as in the mutant cells harboring Ede1 droplets. This suggests that liquid phase separation might be the mechanism through which Ede1 concentrates proteins at the early endocytic sites. We identified the central region of Ede1— containing a coiled-coil and a prion-like domain— as necessary for both the condensate formation, and for Ede1 to promote the initiation of endocytosis. We also found that heterologous prion-like domains can partially replace the Ede1 central region in endocytosis. We also demonstrated that the central region of Ede1 fused to a lipid-binding domain can condense on the plasma membrane. These findings suggest a potential link between endocytic assembly and the phenomenon of protein phase separation and raise questions about the material properties of the endocytic sites at different stages. They also highlight a possible novel role for disordered, prion-like regions found in numerous endocytic proteins (*Malinovska et al., 2013*).

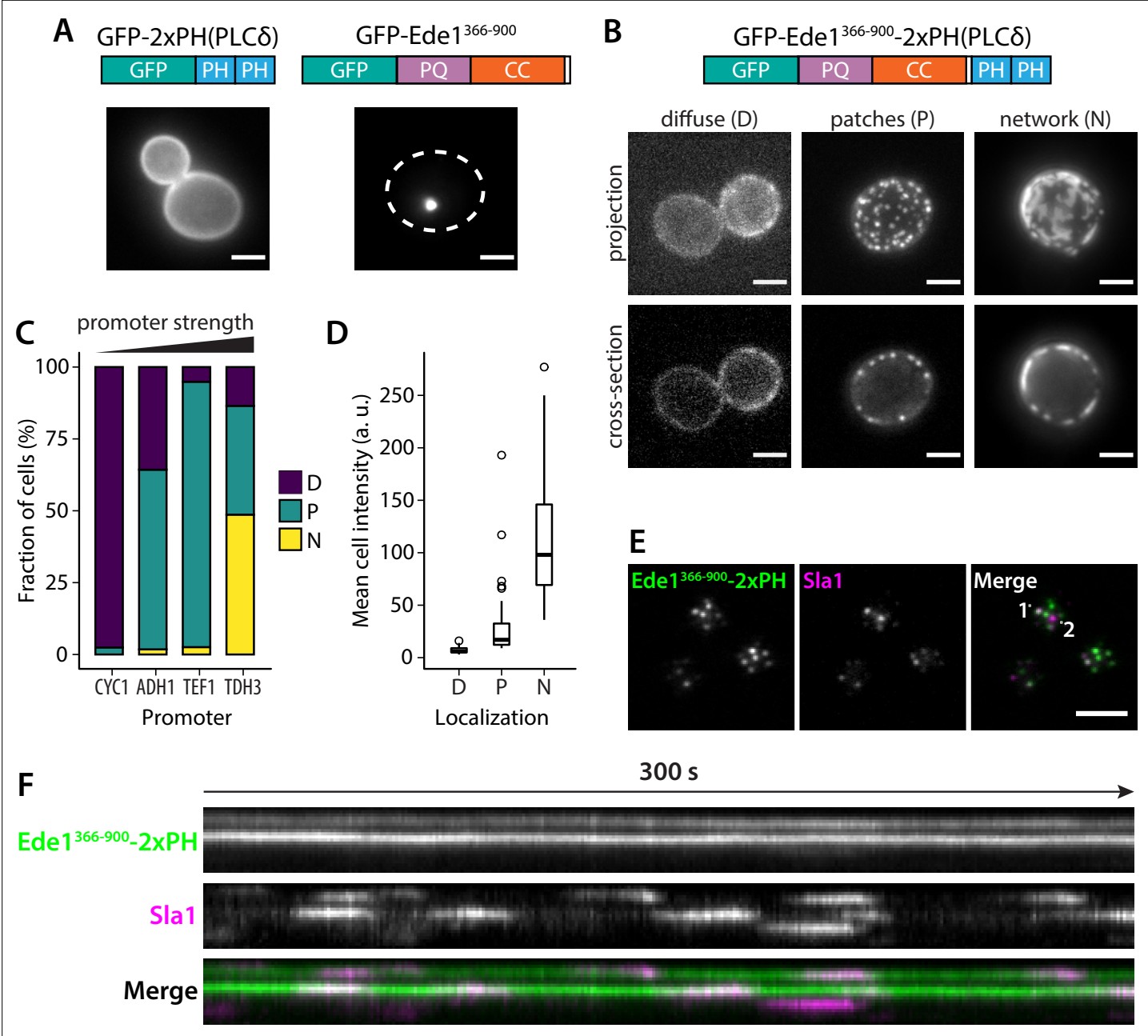

**Figure 9.** Fusion with Ede1 central region changes the distribution of a PI(4,5)P$_2$ probe. (**A**) Maximum projections of cells expressing GFP-2 × PH and GFP-Ede1$^{366-900}$ from a yeast centromeric plasmid under the control of TDH3 promoter. White dotted line shows cell outline. (**B**) Cells expressing GFP-Ede1$^{366-900}$-2×PH from a multicopy plasmid. Examples of different structures classified as 'Diffuse', 'Patches', or 'Networks'. (**C**) and (**D**) The same construct was expressed from four centromeric plasmids under different promoters. Individual cells were classified as in panel B. Plots show percentage of cells falling into each class per promoter (**C**), and mean cell pixel intensity per class (**D**). (**E**) Movies of cells coexpressing GFP-Ede1$^{366-900}$-2×PH and Sla1-mCherry were acquired using TIRF microscopy. Single frame from a representative movie; points labeled '1' and '2' mark the top and bottom of the kymograph (**F**). Scale bars: 2 μm. All cells in this figure: *SLA1-mCherry::KANMX4, ede1Δ::natNT2*.

The online version of this article includes the following video, source data, and figure supplement(s) for figure 9:

**Source data 1.** Source data (panels C and D).

**Figure supplement 1.** Fluorescence recovery after photobleaching and hexanediol treatment of the fusion construct.

**Figure 9—video 1.** The 5-min TIRF movie of GFP-Ede1$^{366-900}$-2×PH (green) and Sla1-mCherry (magenta) represented in panels E and F. https://elifesciences.org/articles/72865/figures#fig9video1

**Figure 9—video 2.** A 3-min equatorial plane movie of GFP-Ede1$^{366-900}$−2×PH (green) and Sla1-mCherry (magenta) showing inward movement of Sla1

*Figure 9 continued on next page*

## Ede1 forms liquid protein droplets

Ede1 forms large condensates under conditions where the stoichiometry between Ede1 and the endocytic adaptor proteins is altered, such as overexpression of Ede1 or deletion of three early adaptors (*Boeke et al., 2014*, *Figure 1*).

We showed that these condensates are liquid, phase-separated droplets according to the following criteria: (a) observation of liquid-like behaviors, (b) molecule turnover, (c) dependence on a critical component concentration, (d) dependence on temperature, and (e) susceptibility to dissolution by 1,6-hexanediol.

We show that the Ede1 condensates undergo apparent fusion and fission events, the latter caused possibly by polymerizing actin filaments (*Figure 4C*, *Figure 4—video 1*). The Ede1 molecules exchange between the condensates and the cytosolic pool (*Figure 2A*) and, importantly, Ede1 molecules also rapidly diffuse within the condensates (*Figure 2C and D*). The formation of the Ede1 condensates depends on the cellular concentration of Ede1 (*Figures 1D and 3*). The condensates dissolve rapidly and reversibly in response to temperature changes (*Figure 2F*), and are sensitive to treatment with 1,6-hexanediol (*Figure 2E*). In agreement with our findings in yeast, *Day et al., 2021* have shown that Eps15, the mammalian homolog of Ede1, can phase separate in vitro.

The Ede1 condensates are clearly distinct from endocytic sites by the virtue of size, brightness, and long-term stability. We assign them no function, other than as a tool used to study the properties of Ede1. However, *Wilfling et al., 2020* studied Ede1 condensates in the 3×ΔEA and Ede1 overexpression strains in parallel to our work. They describe a selective autophagy pathway mediated by Ede1, and propose that autophagy of phase-separated condensates might be a major route through which cells remove misfolded or unneeded endocytic proteins. The ability of Ede1 to cluster endocytic proteins could therefore play a dual role in endocytosis and autophagy.

## Endocytic sites: solid or liquid?

The prevailing model of endocytosis focuses on the growing clathrin lattice as the driver of protein assembly (*Kirchhausen et al., 2014*; *Cocucci et al., 2012*). Indeed, clathrin is a major interaction hub in endocytosis, and a scaffold with a well-defined structure. Nevertheless, several lines of evidence suggest that this model of assembly might be incomplete.

In yeast, the early proteins can assemble in the absence of clathrin. In fact, many endocytic sites in *clc1Δ* cells stall during the early phase, but persist on the plasma membrane (*Carroll et al., 2012*). On the other hand, in the absence of Ede1, numerous early adaptors do not assemble at all, or assemble only for the duration of the membrane-bending phase (*Stimpson et al., 2009*; *Carroll et al., 2012*). These two observations demonstrate that Ede1 can sustain the early sites independently of the clathrin lattice. We hypothesize that Ede1 performs this function by undergoing liquid-liquid phase separation on the plasma membrane.

FRAP shows that numerous proteins involved in endocytosis and the actin cytoskeleton continuously turn over at the endocytic sites (*Skruzny et al., 2012*; *Lacy et al., 2019*). Ede1 is one of such proteins (*Figure 2B*). This observation is consistent with a phase separation mechanism, although it does not prove it. Alternative mechanisms could explain fast turnover, such as 'treadmilling' of actin monomers or dynamic binding of adaptors to the clathrin lattice. Even clathrin itself shows fluorescence recovery as individual triskelia are replaced within the scaffold (*Wu et al., 2001*; *Avinoam et al., 2015*; *Chen et al., 2019*).

Several hallmark criteria frequently associated with liquid phase separation can also be explained by other mechanisms of compartmentalization. This fact has become intensely debated in the context of the nucleus, where bridging of multiple DNA sites could create an appearance of phase-separated compartments (*McSwiggen et al., 2019*; *Peng and Weber, 2019*). Similarly, weak binding to the clathrin lattice could explain the enrichment of proteins at endocytic sites with high turnover and susceptible to dissolution by temperature and 1,6-hexanediol.

Our observations point to a mixed model, in which the structured lattice exists alongside a liquid phase formed by unstructured interactions. The existence of large Ede1 condensates is in itself one of the predictions generated by a phase separation model. A significant consequence of phase separation is that above a critical value, further increase of total component concentration leads to changes in relative volume, but not concentration, of the dense and the light phases. This is in contrast with the scaffold-binding model, where the size of the assemblies formed by endocytic proteins would be limited by the clathrin lattice. In addition, we have observed buffering of cytoplasmic Ede1 concentration during overexpression and in different genetic backgrounds. The cytoplasmic concentration in wild-type cells reached the limit observed in Ede1 overexpression, suggesting that the normal Ede1 concentration is sufficient for its phase separation on the plasma membrane. The small fraction of wild-type cells which contain larger Ede1 condensates (*Figures 1E and 5C*, *Wilfling et al., 2020*) could very well be the consequence of natural variability in expression levels. The concentration buffering also suggests that the phase separation of Ede1 is driven primarily by homotypic interactions (*Riback et al., 2020*).

The qualitative criteria for liquid phase separation—sensitivity to temperature and 1,6-hexanediol—apply to Ede1 at the endocytic sites as well as to the large endocytic condensates. Curiously, the sites appear more stable against both of these treatments than the large condensates. This could be explained by the fact that adaptor binding confines Ede1 at the endocytic sites to the plane of the membrane. As such, the critical concentration could be lower than for the formation of 3D droplets, as shown previously in vitro for the Nephrin/Nck/N-WASP system (*Banjade and Rosen, 2014*). A complementary explanation could be that genuine endocytic sites are stabilized by other interactions, such as those within the clathrin lattice, or the lattice formed by Sla2 and Ent1/2 in the presence of PI(4,5)P$_2$.

The small size of the endocytic sites prevents the direct visualization of liquid-like shape changes. Super-resolution imaging of endocytic proteins in fixed cells revealed that Ede1 and Syp1 form larger and more amorphous structures than clathrin and its adaptors (*Mund et al., 2018*). The super-resolution experiments also suggest that even before it disassembles, Ede1 becomes progressively excluded from the center of the sites. Eps15 and FCHo1 behave in a similar fashion in mammalian cells (*Sochacki et al., 2017*). As the intermediate and late coat proteins form stable patches with low turnover (*Skruzny et al., 2012*), we propose that a liquid-like early module is displaced from the center of the invagination by the formation of a solid coat.

Ede1 is one of the most heavily phosphorylated proteins in yeast endocytosis (*Lu et al., 2016*), and it can be ubiquitylated as well as bind ubiquitin. Phosphorylation can regulate phase separation (*Monahan et al., 2017*; *Larson et al., 2017*), and the phosphorylation of Ede1 might regulate its state at the endocytic sites.

Our chimeric construct GFP-Ede1$^{PQCC}$-2×PH combines a lipid-binding domain with the central region of Ede1. This construct forms bright structures on the surface of the plasma membrane, and the area covered by these structures appears larger in cells with higher expression levels (*Figure 9*). The puncta formed by GFP-Ede1$^{PQCC}$-2×PH repeatedly recruit transient assemblies of the late coat protein Sla1. This suggests that the chimeric construct can initiate functional endocytic events. However, the GFP-Ede1$^{PQCC}$-2×PH puncta persist over long imaging periods and do not disassemble after Sla1 internalization. Structures formed by GFP-Ede1$^{PQCC}$-2×PH also do not recover fluorescence (*Figure 9—figure supplement 1*). This suggests that while the central region of Ede1 can drive condensation, the terminal regions are needed to maintain the liquid state.

It must be noted that we do not fully understand the nature of the microdomains formed by the GFP-Ede1$^{PQCC}$-2×PH construct. For example, all known interaction motifs of Ede1 are located inside the terminal regions. It is thus unclear how the central region could recruit other endocytic proteins.

Prion-like domains are enriched in yeast endocytic proteins (*Alberti et al., 2009*; *Malinovska et al., 2013*), many of which contain polyglutamine tracts longer than that of Ede1. These disordered, low-complexity regions had been previously considered mere linkers (*Dafforn and Smith, 2004*), but we show that the prion-like region of Ede1 is important for its condensation. Our results also show that unrelated prion-like domains can partially replace the function of Ede1 central region, even without the coiled-coil domain (*Figure 8*). How—or if—the endocytic proteins achieve specificity during recruitment of prion-like domains is an open question. It has also been proposed that phase

separation of prion-like domains could provide force for membrane bending (*Bergeron-Sandoval et al., 2021*).

The coiled-coil domain of Ede1 is also critical for its phase separation, and even more important to the function of Ede1 than the prion-like domain (*Figure 6*; *Lu and Drubin, 2017*). The coiled-coil of Eps15 can form dimers and tetramers (*Tebar et al., 1997*; *Cupers et al., 1997*), and fluorescence correlation spectroscopy (FCS) data suggests that cytosolic Ede1 can form dimers and higher-order oligomers (*Boeke et al., 2014*). Multivalency is a critical characteristic of phase-separating proteins (*Li et al., 2012*; *Banani et al., 2016*), and oligomerization via the coiled-coil domain could promote phase separation by increasing the valency of other interactions. Less commonly, coiled-coils can also form phase-separating networks in absence of other interaction motifs, such as in the case of centrosome scaffold SPD-5 (*Woodruff et al., 2017*). In our experiments, replacing the central region of Ede1 with heterologous coiled-coils partially rescued Ede1 localization, but not the late phase defect, of *ede1^ΔPQCC* cells.

## Materials and methods

### Yeast strains and plasmids

The list of yeast strains and yeast plasmids used in this study is provided in *Supplementary file 1*. These materials are available upon request to the corresponding author.

Cells were maintained on rich medium at 24 or 30°C. C-terminally tagged or truncated mutants were generated via homologous recombination with PCR cassettes as described by *Janke et al., 2004*. N-terminal truncation and internal domain deletion or replacement mutants of Ede1 were generated by first constructing the desired mutant gene in a pET-based plasmid using PCR mutagenesis or ligation-independent cloning (*Li et al., 2012*). The mutated Ede1 sequence was then amplified by PCR using primers containing 50 bp overlap with 5' (forward primer) 3' (reverse primer) UTR sequences of EDE1. The PCR product was transformed into *ede1Δ::klURA3* cells. The transformants were selected for on plates containing 5-fluorootic acid, and confirmed by colony PCR and genomic sequencing.

Sequences coding for IDRs replacing Ede1 core region in *Figure 8* were obtained from several different sources. The coding sequences of Snf5 and Whi3 IDRs were amplified by PCR from the yeast genome and were confirmed identical to the sequences in the S288C reference genome (SGD:S000000493 and SGD:S000005141, respectively). The coding sequence of amino acids 1–253 of the Sup35NM3 mutant (*Franzmann et al., 2018*) was cloned from a plasmid kindly provided by Titus Franzmann. The coding sequence of human FUS low-complexity region (amino acids 2–214 from UniProt entry Q6IBQ5) was codon optimized for yeast expression, and synthesized with the rest of the Ede1 sequence by Synbio Technologies (New Jersey). The sequence coding for amino acids 335–931 of *D. melanogaster* kinesin-1 was cloned from Addgene plasmid K980 (#129762), a gift from William Hancock. The sequence coding for amino acids 358–797 of *Homo sapiens* kinesin-5 was cloned from Addgene plasmid mCherry-Kinesin11-N-18 (#55067), a gift from Michael Davidson.

Plasmids used in *Figure 9* were based on pRS426-GFP-2×PH(PLCδ) (*Stefan et al., 2002*), a kind gift from Scott Emr. The Ede1^366-900 coding sequence was inserted into this plasmid after the last GFP codon using ligation-independent cloning. GFP-2×PH(PLCδ) and GFP-Ede1^366-900-2×PH(PLCδ) were then subcloned to a pRS416-based plasmid p416-GPD under the control of TDH3 promoter (*Mumberg et al., 1995*) using BamHI and SalI restriction sites. GFP-Ede1^366-900-2×PH(PLCδ) was also subcloned into p416-CYC1, p416-ADH1, and p416-TEF1 (*Mumberg et al., 1995*) using the same restriction sites.

The sequence of the msGFP2 fluorophore (*Valbuena et al., 2020*) was cloned into a PFA6a-based tagging plasmid from Addgene plasmid #135301, a kind gift from Benjamin Glick.

### Live cell imaging

Yeast cells were grown to $OD_{600}$ between 0.3 and 0.8 at 24°C in low-fluorescence synthetic drop-out medium lacking tryptophan, or tryptophan and uracil if required for plasmid maintenance. Cells were attached to cover slips coated with 1 mg ml$^{-1}$ concanavalin A.

## Widefield microscopy

Widefield micrographs were obtained on an Olympus IX81 widefield microscope equipped with a 100×/NA1.45 objective and an ORCA-ER CCD camera (Hamamatsu), using an X-CITE 120 PC (EXFO) metal halide lamp as the illumination source. The excitation and emission light when imaging EGFP- and mCherry-tagged proteins were filtered through the U-MGFPHQ and U-MRFPHQ filter sets (Olympus). The 3D stacks were acquired with 0.2 μm vertical spacing. The microscope was controlled using the MetaMorph software (Molecular Dynamics).

## Total internal reflection fluorescence microscopy

All TIRF movies were recorded on an Olympus IX83 widefield microscope equipped with a 150×/NA1.45 objective and an ImageEM X2 EM-CCD camera (Hamamatsu) under the control of the VisiView software (Visitron Systems). The 488 nm and 561 nm laser lines were used for illumination of GFP- and mCherry-tagged proteins. Excitation and emission were filtered using a TRF89902 405/488/561/647 nm quad-band filter set (Chroma). Laser angles were controlled by iLas2 (Roper Scientific).

## Fluorescence recovery after photobleaching

Bleaching of Ede1-EGFP in endocytic condensates (*Figure 2A and B*) was performed using a custom-built set-up that focuses a 488-nm laser beam at the sample plane, on the Olympus IX81 widefield microscope described above. The diameter of the bleach spot was approximately 0.5 μm.

Bleaching of unperturbed endocytic sites (*Figure 2C*) was performed with a 405 nm laser line controlled by the iLas2 targeting system during simultaneous excitation with 488 nm and 561 nm lasers in TIRF mode on the Olympus IX83 microscope described above. The emission light was collected through a Gemini beam splitter (Hamamatsu) equipped with a Di03-R488/561-t1 dichroic, and FF03-525/50-25 and FF01-630/92-25 emission filters (Semrock).

## Spinning disk microscopy

Spinning disk confocal imaging (*Figure 3*) was performed in the Photonic Bioimaging Center at the University of Geneva using a Nikon Eclipse Ti1 microscope equipped with a CSU-W1 spinning disk (Yokogawa) using a 100×/NA1.49 objective, an sCMOS Prime 95B camera (Photometrics), and 488 nm and 561 nm lasers as the illumination source.

## Induction of protein expression

For the induction of expression from GALS promoter during live-cell imaging (*Figure 3*), cells were thawed and grown for several days on Synthetic Complete medium agar plates with 2% galactose as the sole carbon source. The cells were then cultured overnight in a low-fluorescence synthetic drop-out liquid medium with no tryptophan and 2% raffinose as the sole carbon source. The cells were diluted in the morning into the same medium with 2% glucose as the carbon source. The cells were attached to cover slips as described above. Finally, the carbon source in the medium was switched to 2% galactose before the start of the imaging.

## **Image and data analysis**

All code used in this study is available as a single repository at https://github.com/matkozak/KozakAnd-Kaksonen2022, (copy archived at swh:1:rev:5441acf218619f2b03d90633613cccc373c6fe8a; *Kozak, 2022*). General image analysis was performed using the Fiji distribution of ImageJ (*Schindelin et al., 2012*; *Rueden et al., 2017*). All display images were corrected for background fluorescence using the rolling ball algorithm of ImageJ, and movies were corrected for photobleaching using a custo m ImageJ macro. Plots and statistical analyses were generated using R.

## FRAP experiments

FRAP experiments performed on Ede1-EGFP condensates were analyzed according to *Phair et al., 2004*. Mean fluorescence values were measured from regions of interest representing the background, the cell, and the condensate. A custom-written R script (available in the article repository) was used to subtract background fluorescence, correct for photobleaching and normalize the values between 0 (corrected fluorescence immediately after photobleaching) and 1 (mean corrected fluorescence of

5 s before photobleaching). The recovery curves of individual experiments were aligned to bleach time and averaged. Condensates that showed lateral or axial movement during the acquisition were manually excluded from the averaging. The average was fitted to a single exponential equation from which the mobile fraction and recovery half-time were calculated.

For FRAP experiments performed on native endocytic sites, the background fluorescence was first subtracted from the TIRF images using the ImageJ rolling ball algorithm. EGFP and mCherry fluorescence of single endocytic patches were measured within a circle with a radius of three pixels around the patch centroid position. A custom-written R script was used to calculate the fluorescence recovery much in the same way as for the FRAP of condensates, but no further corrections were made for background signal or imaging-induced photobleaching. To calculate average recovery, we manually selected only events in which Abp1-mCherry signal peaked at least 60 s after bleach time to exclude the effect of Ede1 disassembly at the end of endocytic events.

## Patch numbers and lifetimes

For estimating the number of patches per membrane area, we analyzed single nonbudding cells. The patches were thresholded and counted using a custom Python script available in the article repository. We estimated the cell surface area by measuring the area of the cross-section from maximum intensity projection and multiplying it by 4, under the assumption that an unbudded yeast cell is approximately spherical. For estimating patch lifetimes, we tracked endocytic events using ParticleTracker from the MOSAIC suite (*Sbalzarini and Koumoutsakos, 2005*) and multiplied trajectory length by the frame rate.

## Cytosolic and total cellular intensity

To obtain cytosolic intensity of Ede1-EGFP, 5 × 5-pixel square regions away from the condensates and vacuoles were manually measured in ImageJ. To measure total cellular intensity, individual cells were cropped in ImageJ. A custom Python script was applied to the cropped cells to generate masks based on Rvs167-mCherry fluorescence, and subsequently measure the Ede1-EGFP signal intensity in the masked region.

## Cell classification

To calculate the percentages of Ede1-EGFP condensates colocalizing with mCherry puncta in *Figure 4*, cells containing Ede1-EGFP condensates in *Figure 5* and cells showing different GFP-EDE1$^{366-900}$–2×PH localization patterns in *Figure 9*, single cells were cropped from imaging fields based on a neutral signal (GFP in the case of *Figure 4* and brightfield image for *Figures 5 and 9*). Next, an ImageJ macro was used to display random images from the dataset and the experimenter would assess the presence of the tested phenotype with no knowledge of which strain was being analyzed.

## Western blotting

A 300 µl of ice-cold trichloroacetic acid was added to 5 ml of exponentially growing yeast cultures. The cells were pelleted by centrifugation, washed with cold acetone, and dried in a vacuum concentrator. The pellets were resuspended in 100 µl of urea buffer (25 mM Tris-HCl pH 6.8, 6 M urea, 1% SDS) and homogenized by shaking with 200 µl of glass beads. The samples were heated at 95°C for 5 min, mixed with 100 µl 2× SDS loading buffer and centrifuged at 16, 000× g for 5 min.

The samples were subjected to electrophoresis on 4–20% Precast Protein Gels (Bio-Rad) and transferred onto a nitrocellulose membrane using an iBlot2 device (ThermoFischer Scientific). The membranes were blocked for 30 min with 5% bovine serum albumin in PBS-Tween and incubated with primary antibodies overnight at 4°C. The membranes were washed in PBS-Tween, incubated with fluorescent secondary antibodies for 1 hr and washed in PBS-Tween. The fluorescence was measured on an Odyssey scanner (LI-COR Biosciences).

## Antibodies

Ede1 constructs were detected using an anti-GFP mouse monoclonal antibody (ab291, Abcam) at 1/2000 dilution, and an anti-Hog1 rabbit polyclonal antibody (sc-9079, Santa Cruz Biotechnology) at 1/1000 dilution was used as a loading control. Donkey antimouse IRDye 680 and anti-rabbit IRDye

800 secondary antibodies (926–68072 and 926–32213 respectively, LI-COR Biosciences) were used at a 1/10,000 dilution.

## Acknowledgements

This work was supported by the Swiss National Science Foundation (grants 31003A_163267 and 310030B_182825) and by the NCCR Chemical Biology funded by the SNSF.

We are thankful to all the members of the Kaksonen laboratory, especially Markus Mund, Andrea Picco, and Daniel Hummel for their critical reading of the manuscript. We also thank Jeanne Stachowiak and Kasey Day for their comments and Camilla Godlee for contributions to the early phase of the project.

## Additional information

### Funding

| Funder | Grant reference number | Author |
| --- | --- | --- |
| Swiss National Science Foundation | 31003A_163267 | Marko Kaksonen |
| Swiss National Science Foundation | 310030B_182825 | Marko Kaksonen |
| NCCR Chemical Biology | | Marko Kaksonen |

The funders had no role in study design, data collection and interpretation, or the decision to submit the work for publication.

### Author contributions

Mateusz Kozak, Conceptualization, Data curation, Formal analysis, Investigation, Methodology, Software, Validation, Visualization, Writing - original draft, Writing - review and editing; Marko Kaksonen, Conceptualization, Funding acquisition, Writing - original draft, Writing - review and editing

### Author ORCIDs

Mateusz Kozak (iD) http://orcid.org/0000-0002-1354-693X
Marko Kaksonen (iD) http://orcid.org/0000-0003-3645-7689

### Decision letter and Author response

Decision letter https://doi.org/10.7554/eLife.72865.sa1
Author response https://doi.org/10.7554/eLife.72865.sa2

## Additional files

### Supplementary files

- Transparent reporting form
- Supplementary file 1. Yeast strains and plasmids used in the study.

### Data availability

All data generated or analysed during this study are included in the manuscript and supporting source data files.

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
