## [Editor Report]

This article demonstrates that the early arriving endocytic protein Ede1, the yeast Eps15 homolog, can generate a separate liquid-phase in vivo when overexpressed and that the domains conferring this property are required and sufficient to nucleate endocytic patches. The genetic and microscopy results are compelling and they support the model of liquid-phase separation in endocytosis, even though alternatives are discussed.

---

## [Decision Letter]

**Decision letter after peer review:**

[Editors’ note: the authors submitted for reconsideration following the decision after peer review. What follows is the decision letter after the first round of review.]

Thank you for submitting your paper entitled "Phase separation of Ede1 promotes the initiation of endocytic events" for consideration by *eLife*. Your article has been reviewed by 3 peer reviewers, including Maria Isabel Geli as the Reviewing Editor and Reviewer #1, and the evaluation has been overseen by Vivek Malhotra as senior editor. The following individual involved in review of your submission has agreed to reveal their identity: Stephen J Royle (Reviewer #2).

We are sorry to inform you that at this point the manuscript cannot be considered for publication in *eLife*.

In summary, there is a consensus on the importance of the scientific question addressed by your work, and on the nature of the Ede1 condensates when it is overexpressed or in the endocytic adaptor mutant background. However, 2 of 3 reviewers considered that the manuscript does not demonstrates that Ede1 really undergoes liquid-liquid phase separation at endocytic sites under physiological conditions and whether this property is required for its endocytic function. The concerns follow.

1. Several of the key experiments aimed at showing phase separation yield different results for the condensates and the endocytic sites. Basically, they disassemble at different temperatures and at different hexanediol concentrations. Together with the fact that FRAP per se does not demonstrate that Ede1 forms a separate liquid phase at endocytic sites, the data rather suggest that the Ede1 condensates and the Ede1 endocytic patches have different properties.

2. The experiment showing that domains required to form the condensates are required for endocytosis is not really conclusive because, among other considerations, there seems to be an inverse correlation between the capacity of the different mutants to be recruited at endocytic sites and their capacity to sustain endocytosis. Under these circumstances, the endocytic defect observed in the ∆PQCC mutant could be due to the absence of the N or the C-terminal portions of Ede1 at endocytic sites, rather than the absence of polyQ and coiled-coil domains.

Demonstrating conclusively that Ede1 undergoes a phase transition at endocytic sites and this property is required for endocytic uptake is challenging and will take considerable effort and time. But in essence, these are the points that would actually move the field forward, since phase transitions have been now demonstrated for many proteins, but its functional relevance in many contexts including endocytosis, remains elusive.

We list here a few experiments that could help you prepare a stronger manuscript, but there are likely many more. (1) to hook the wild type Ede1 or the ∆PQCC mutant to an early endocytic protein (i.e. Syp1) in an ede1∆ background and show that the strain expressing the mutant but not the wild type still has an endocytic defect; (2) to finely titrate the concentration of cytosolic Ede1 from 0 to physiological levels and show that the concentration of Ede1 at endocytic sites is sigmoidal rather than linear; (3) to show that another IDR capable of undergoing liquid-phase separation can substitute for the PQ and CC domains of Ede1; or (4) to demonstrate that the PH-PQ-CC construct can recruit the actin machinery and trigger actual endocytic events, but that this capacity is lost under experimental conditions that dissolve the Ede1 condensates (5 min 37oC or 5% hexanediol).

*Reviewer #1:*

The manuscript addresses what has been a debated issue in the last years regarding the role of intrinsic disordered domains and liquid-liquid phase separation in endocytosis. While it is clear that many endocytic proteins including for example AP180 or Epsins contain this kind of domains, demonstration of a role of liquid-liquid phase separation in the context of membrane budding in vivo has remained elusive. In this work, Kozak and Kaksonen identify a region in Ede1, an early endocytic factor of the yeast *S. cerevisiae*, which seems to be necessary and sufficient to trigger liquid-liquid phase separation when overexpressed in vivo (based on 4 criteria: the observation of fission and fusion events, fast recovery upon photobleaching, reversible disassembly upon heating and sensitivity to 1,6-hexanediol). Deletion of this region in Ede1 causes endocytic defects, whereas its covalent attachment to a PH domain, which targets it to the plasma membrane, generates cortical patches that recruit other endocytic components and seem to trigger endocytic events.

While the data is interesting, it is still a bit preliminary, and it does not really probe at the moment that the putative liquid-liquid-phase separation induced by this Ede1 region is the relevant feature required for endocytic uptake. The authors would need to take into consideration a few points:

The authors show that the Ede1 foci undergo fusion and fission events and use this observation to support that the Ede1 foci correspond to a separated liquid phase. How often are these events observed? Are those fusion and fission events associated to membranes?

Does Ede1 undergo liquid-liquid phase separation in vitro? The authors mention that this is the case for the mammalian homolog Eps15, but they would need to show that this is the case for Ede1.

Other endocytic proteins have intrinsically disordered domains. How is temperature and 1,6-hexanediol affecting these proteins. Could the authors include controls of other bona-fide non-membrane bound organelles for comparison?

The authors show that deletion of the Ede1 region inducing liquid-liquid phase separation install endocytic defects, but the corresponding construct is not recruited to endocytic sites, so the defects could also be caused by the absence of other Ede1 domains. The authors would need to artificially hook the Ede1 mutant to another early component (i.e. Syp1) and check then if an endocytic defect is installed.

Are the effects of the PH-Ede1 construct, and in particular Sla1 recruitment, especially sensitive to temperature and/or 1,6-hexanediol treatment? It is difficult to evaluate from the kymographs in figure 7 if Sla1 fades away or it is internalized. Can the authors show images of wild type Ede1/Sla1 patches and quantify internalization in both situations? Is Abp1 also recruited to the PH-Ede1 foci? This would be expected if internalization occurs.

*Reviewer #2:*

Kozak and Kaksonen propose that Ede1 may initiate endocytic events due to its phase separation properties. We found this manuscript very interesting. There's no escaping phase separation right now in cell biology, but this manuscript strikes a nice balance between hype and reality because they are upfront about what is studied here (by overexpression or expression in mutant background) and what it suggests may be happening in normal cells during endocytosis. We think the experiments presented do support the conclusions. Complementary work on bioRxiv shows similar behaviour of the mammalian homologue Eps15, which gives confidence that the authors are on the right track here.

There was one major point that if addressed experimentally (under non-pandemic circumstances) would strengthen the manuscript. The data on the condensates are convincing, but we wondered whether the Ede1 blobs are truly liquid condensates that are devoid of membrane. In our own work in mammalian cells we have seen large blobs of overexpressed protein that turn out to be MVB-like structures by EM. These structures show similar FRAP profiles to the data here, and I don't think any of the experimerts presented here necessarily rule out that these structures may have a membrane component.

Our remaining points are straightforward to address with existing data and would help to improve the paper.

1. Figure 1D number of cells with condensates is annotated. Are there differences in the number or brightness of condensates between EDE1-EGFP/EDE1-EGFP and EDE1-EGFP/EDE1?

2. Figure 2C would be improved by including the quantification that is listed in the text i.e. graph with average and fit.

3. Figure 3 who recruits who in this figure? It is convincing that the blob is due to Ede1 but the site of the blob could be dictated by any one of the other proteins. Do movies of blob formation show that each protein appears after Ede1 or do they simultaneously accumulate?

4. Figure 4C shows FL gives 100% of cells with condensates but the percentage is lower in Figure 1D – is this just experimental variability?

5. Figure 5CD is there a correlation between patch density and lifetime? If they are plotted against each other do they scale linearly? Also, the movies in the manuscript work well and for this figure, a side-by-side movie of wt, ∆PQCC and ede1∆ would be a useful addition.

6. Figure 6 shows a representative cell. Quantification would strengthen this figure.

*Reviewer #3:*

In this paper, Kozak and Kaksonen report that the budding yeast's early endocytic protein Ede1, a homologue of mammalian Esp15 can form aggregates when the protein is overexpressed or expressed in absence of 3 other early endocytic proteins (the AP180 family proteins Yap1801/2 and the AP2 complex α-subunit Apl3). The paper aims at showing that these aggregates are phase separated structures and speculate that phase separation could be the mode of assembly of Ede1 during clathrin-mediated endocytosis in yeast. The authors also identified the minimum portion of Ede1 that leads to these aggregates.

1. The data are relatively solid to demonstrate that the aggregates seen in the mutants can be phase separated structures. However, I am not convinced the data strongly support that Ede1 experiences phase separation at sites of clathrin-mediated endocytosis. To my opinion, several pieces of data strongly suggest that Ede1 assembles into two different types of structures in the mutant strains – one structure at the endocytic sites, which is due to specific high affinity interactions, and one structure at the aggregates that is partly due to new low affinity interactions that are not present in normal conditions at the endocytic sites (possibly in addition to the other high affinity interactions mentioned above). Indeed, several of the key experiments that aim to show phase separation yield different results for the aggregates and the endocytic sites:

(1a) Aggregates and endocytic sites are disassembled at different temperature. This is expected from the laws of thermodynamics if the interactions in aggregates and endocytic sites have different affinities. Indeed, the Kd of any reversible biochemical interaction depends exponentially on the temperature and low affinity interactions (from the aggregates) will become unfavorable at lower temperature than higher affinity interactions (from the endocytic sites).

(1b) 1,6-hexanediol dissociates aggregates at lower concentrations than it dissociates endocytic patches, which also suggest the interactions in the aggregates have lower affinity. As a side note, I want to point out that even though researcher in the phase separation field routinely use hexanediol to demonstrate phase separation, it remains somewhat controversial [McSwiggen et al. 2019].

(1c) FRAP on endocytic sites does not demonstrate that they are phase separated structures: many other endocytic proteins do exchange rapidly and are not believed to be phase separated (e.g. the actin cytoskeleton proteins [Kaksonen et al. 2003, Kaksonen et al. 2005, Lacy et al. 2019]).

(1d) The mobile and immobile fractions are different in aggregates and endocytic sites.

2. Since Ede1 concentration seems to be critical for the formation of the aggregates, it is important to systematically report the cytosolic and cell concentrations (or even the relative intensities) in different experimental conditions:

(2a) l. 100, it is suggested that the levels of cytoplasmic Ede1 might be different in wild-type and the triple mutant. Concentrations (even relative) should be reported.

(2b) In Figure 4 it is important to make sure the aggregation with different constructs is not due to differences in the cytoplasmic or cell concentrations of the constructs.

(2c) Figure 7C and l. 252+: how the behavior of the puncta formed with the Ede1(366-900)-2xPH construct depend on concentration? Also, do all the puncta recruit Sla1? Is Sla1 also recruited at sites where the Ede1 construct is not present?

3. l. 157: I am not sure I understand the logic (I may have missed something). The fact that it is a condensate does not imply it triggers actin polymerization. This is a correlation, not a causation. In addition, one could imagine there is a defined endocytic patch within a diffraction limited distance from the condensate. Quantifying the amount of abp1 (relatively to other patches) could be a way to tell whether these are bona fide endocytic structures coming out of the aggregate.

4. In conclusion, in its current form, this study does convince me that Ede1 phase separates during clathrin-mediated endocytosis. In addition, it is unclear to me how the putative phase separation of Ede1 would affect endocytosis, compared to regular protein assembly. I believe several extra experiments would be necessary to make a strong point.

[Editors’ note: what follows is the authors’ response to the second round of review.]

Thank you for submitting your article "Phase separation of Ede1 promotes the initiation of endocytosis" for consideration by *eLife*. Your article has been reviewed by 3 peer reviewers, and the evaluation has been overseen by a Reviewing Editor and Anna Akhmanova as the Senior Editor. The following individuals involved in review of your submission have agreed to reveal their identity: Stephen J Royle (Reviewer #2).

The manuscript addresses what has been a debated issue in the last years, regarding the role of intrinsic disordered regions (IDRs) and liquid phase separation in endocytosis. While it is clear that many endocytic proteins, including AP180 or Epsins, contain this kind of sequences, conclusive demonstration of a role of liquid phase separation in the context of membrane budding in vivo has remained elusive. In this work, Kozak and Kaksonen identify a region in the yeast Eps15 homolog Ede1 (comprising a coiled-coil domain (CC) and a PQ rich region (PQ)), which is necessary and sufficient to trigger liquid phase separation when overexpressed in vivo (based on 4 criteria: the observation of fission and fusion events, fast recovery upon photo-bleaching, reversible disassembly upon heating and sensitivity to 1,6-hexanediol). Deletion of this region in Ede1 causes endocytic defects, whereas its covalent attachment to a PH domain, which targets it to the plasma membrane, generates cortical patches that recruit other endocytic components. Strikingly also, IDRs from other unrelated proteins known to trigger liquid phase separation can partially complement the Ede1 mutant lacking the CC and PQ regions.

While the data is of high quality and consistent with the role of liquid phase separation in endocytosis, other models might as well explain the data. The ectopic IDRs might work as linkers or spacers of the N and C-terminal domains, or they might promote oligomerization. In the context of this debated hypothesis, the other possibilities need to be tested. Also, the requirement for the CC and PQ regions in endocytosis is not conclusive because the mutant lacking these domains is not recruited to endocytic sites, so the endocytic defect might as well be caused by the absence of the N or C-terminal domains.

After some discussion on the new version of the manuscript, there is a general agreement on the interest of the article and the quality of the data. However, two out of three reviewers still consider that the functional relevance of liquid phase separation in endocytosis is not conclusively proven and other hypothesis might as well explain the data. Even though the observation that other unrelated Intrinsically Disorder Regions (IDRs) can partially substitute for the Ede1 regions promoting liquid phase separation is quite compelling, the inserted IDRs might as well work by providing a flexible loop of a given size between the N and C terminal domains or by promoting oligomerization. Since the role of liquid phase separation in endocytosis is under strong debate, we feel that these two other possibilities need to be properly tested and discussed before publication. Even if the new data would disprove the liquid phase separation model, the article would still be of interest for *eLife*. If the new oligomerization or linker regions cannot functionally substitute for the Ede1 coiled-coil (CC) and PQ rich (PQ) regions, the data would provide stronger support for the role of liquid phase separation in endocytosis but still, the title, the abstract and the discussion would need to be rephrased to avoid overstatements that would slow down advance in the field. In addition to this main concern, there is still the issue on the recruitment of the Ede1-(PQ,CC)∆ mutant to endocytic sites. At this point, it is not possible to conclude that the CC-PQ region is required for endocytic uptake because the mutant is not recruited to endocytic sites. One of the reviewers also suggested to investigate the possible differences in the posttranslational modifications that could lead to phase-separation. Even though this approach could be enlightening, it will take more than two months and it is probably beyond the scope of the manuscript. In essence then, the manuscript would be adequate for publication in *eLife*, provided than you address the following key points:

1. Substitute the CC and PQ regions for non-IDRs that can act as linkers or oligomerization domains and investigate if the chimeras can complement the ede1∆ phenotype.

2. Solve the issue of the recruitment of the Ede1-(PQ,CC)∆ to endocytic sites.

3. Rewrite the title, the abstract and the discussion to make clear that the results support the requirement for liquid-phase separation in endocytosis but they do not conclusively prove it.

In addition to these main concerns, please carefully read the rest of the points raised by the reviewers and discuss or address them:

1. The new data showing Ede1 concentration buffering is interesting. However, we find the authors' interpretation unclear at places and, to our understanding, only one possible interpretation of the data. The authors say the cells control their cytoplasmic concentration of Ede1 but, to them, is it an active or a passive process? For bona fide purely phase separating proteins, this can be a passive process (once phase separation happens, the soluble concentration remains virtually constant and the size of the condensates changes with changing concentration). The data presented in the paper is indeed compatible with this hypothesis. However, the data is also compatible with equally likely alternative hypotheses. One is that there are two populations of Ede1 with different post-translational modifications because of the overexpression. As the authors point out, Ede1 is the most heavily phosphorylated protein in CME (Lu et al. 2016), and it is also ubiquitinated. Therefore, it is possible that overexpressing Ede1 overloads its kinases, phosphatases and/or ubiquitination machineries, and the excess mis-phosphorylated/mi-ubiquitinated proteins would phase separate, without being involved in endocytosis. It is also possible that Ede1 does not normally phase separate at endocytosis sites because it is bound to endocytic partners or cargo. Overexpressing Ede1 would create a stoichiometric imbalance at endocytic sites and the excess protein would end up in the cytoplasm and could phase separate there only.

2. We find it surprising that the FUS domain does not phase separate Ede1 since FUS is a well known phase separating domain, and other phase separating domains seem to work. Do the authors have an explanation for that?

3. L 208: "catalyzes cycles of assembly and disassembly of actin": it is not clearly demonstrated, so this could reformulated.

4. L 392: "likely": change to "possibly" because this is not demonstrated in the paper (just correlation with abp1).

5. In response to reviewer #2, the authors point out that hexanediol causes the PH domain to form clusters, but say they are not showing it in the paper because it is only peripherally related to the paper. I agree but this could be useful for the field to report this anyway because there are a lot of controversies about the use of hexanediol in the phase separation field.

6. Talking about fusion and fission of condensates is an overstatement. It might just be the condensates getting closer or separating. Talking about apparent fission or fusion events throughout the text would be more accurate. How often are the putative fission events observed?

7. Is the PQ region alone generating condensates? The authors test the 1-591 Ede1 region containing the EH domains and the PQ region, but not the PQ region alone. The EH domain might somehow inhibit the formation of condensates by the IDR.

8. Are the condensates co-localizing FM4-64 stainable membranes? If so, the membrane association might provide for the liquid-phase-like behavior.

9. Does Abp1 assemble on all condensates on the time-lapse movies? Does the transient Abp1 assembly on the condensates explain the low percentage of co-localization? Are condensates hot spots for Abp1 assembly?

10. Are the GFP-Ede1(366-900) condensates behaving as the GFP-Ede1 condensates in terms of temperature and hexanodiol sensitivity? It is surprising that the GFP- Ede1(366-900)-2xPH condensates do not. If the GFP-Ede1(366-900) condensates behave as separate liquid-phase condensates, similar to those generated by Ede1, it might mean that their membrane association modifies the Ede1-liquid-phase properties. In this context, are the PM associated Ede1 condensates (dome-shaped) behaving similar to those assembled within the cytosol?

11. It is still not so clear if the GFP- Ede1(366-900)-2xPH can really initiate bona fide endocytic events. The internalization of the Sla1 patches is not clear in the images provided. If they are bona fide endocytic events, the fact that GFP- Ede1(366-900)-2xPH condensates do not behave as a liquid-phase would suggest that the formation of an Ede1-dependent liquid phase is not really essential for endocytic uptake and that the Ede1 central region rather operates as a linker or by recruiting other endocytic proteins.

12. The Rapamycin inducible system might not be the most adequate to bring the Ede1 trucations to endocytic sites because the tor1 mutations used to make the yeast strain resistant to Rapamycin might interfere with endocytic uptake. Are Syp1 and Sla1 cortical patch dynamics similar to the wild type´s in a tor1-1 frp1 EDE1-mCherry-FKPB SYP1-FRB background? It might be easier to covalently link the Ede1 truncations to Syp1 or to a later endocytic coat component that is less affected by depletion of Ede1.

*Reviewer #1:*

Kozak and Kaksonen demonstrate that the mammalian Eps15 yeast homolog Ede1 can phase-separate upon overexpression or deletion of endocytic adaptors in vivo. The authors define the protein regions necessary and sufficient to generate the condensates and demonstrate that those are probably necessary and sufficient to sustain the endocytic function of Ede1. Conclusive demonstration on whether liquid phase-separation actually occurs at endocytic sites in wild type cells remains elusive. However, the authors show that unrelated protein sequences known to undergo this transition can functionally (albeit partially) substitute for the Ede1 region generating the condensates.

Even though the data points to the functional role of a separated Ede1 liquid phase in endocytic uptake, some important issues are still unsolved:

1. The authors conclude that the coiled-coil and IDR of Ede1 is required for endocytic uptake because using a Rapamycin inducible system to hook the Ede1 mutant lacking these domains to Syp1, does not complement the ede1∆ defects. Again, this particular experiment is not very informative because the Ede1 mutant is not really recruited to endocytic sites in the presence of Rapamycin and therefore, the endocytic defect might be because by the absence of the N or C-terminal domains. The Rapamycin inducible system might not be the most adequate to bring the Ede1 trucations to endocytic sites because the tor1 mutations used to make the yeast strain resistant to Rapamycin might interfere with endocytic uptake. Are Syp1 and Sla1 cortical patch dynamics similar to the wild type´s in a tor1-1 frp1 EDE1-mCherry-FKPB SYP1-FRB background? It might be easier to covalently link the Ede1 truncations to Syp1.

Other important but easily addressable points:

1. Talking about fusion and fission of condensates is an overstatement. It might just be the condensates getting closer or separating. Talking about apparent fission or fusion events throughout the text would be more accurate. How often are the putative fission events observed?

2. Is the PQ region alone generating condensates? The authors test the 1-591 Ede1 region containing the EH domains and the PQ region, but not the PQ region alone. The EH domain might somehow inhibit the formation of condensates by the IDR.

3. Are the condensates co-localizing FM4-64 stainable membranes? If so, the membrane association might provide for the liquid-phase-like behavior.

4. Does Abp1 assemble on all condensates on the time-lapse movies? Does the transient Abp1 assembly on the condensates explain the low percentage of co-localization? Are condensates hot spots for Abp1 assembly?

5. Are the GFP-Ede1366-900 condensates behaving as the GFP-Ede1 condensates in terms of temperature and hexanodiol sensitivity? It is surprising that the GFP- Ede1366-900-2xPH condensates do not. If the GFP-Ede1366-900 condensates behave as separate liquid-phase condensates, similar to those generated by Ede1, it might mean that their membrane association modifies the Ede1-liquid-phase properties. In this context, are the PM associated Ede1 condensates (dome-shaped) behaving similar to those assembled within the cytosol?

6. It is still not so clear if the GFP- Ede1366-900-2xPH can really initiate bona fide endocytic events. The internalization of the Sla1 patches is not clear in the images provided. If they are bona fide endocytic events, the fact that GFP- Ede1366-900-2xPH condensates do not behave as a liquid-phase would suggest that the formation of an Ede1-dependent liquid phase is not really essential for endocytic uptake and that the Ede1 central region rather operates as a linker or by recruiting other endocytic proteins.

*Reviewer #2:*

The authors have addressed all the points that I raised in my original review. I feel that the new work added has strengthened the manuscript.

During the last review, I was quite positive about this paper; the other reviewers and editors less so. I share the general scepticism about "phase separation explains all cell biology" hype, however I think this is a careful study and the authors are quite conservative in their interpretation.

Since this manuscript was posted it has been cited 10 times (according to Google Scholar today) suggesting the community are finding it useful. I also note the companion preprint by Day et al. (showing similar results for the human homolog) was published in Nature Cell Biology back in April. I don't think delaying publication of this manuscript further is helping anyone.

*Reviewer #3:*

We thank the authors for their efforts in addressing our initial concerns and providing new data. We think the paper has been improved since the first submission. However, we think there are still important remaining issues.

1. The authors provide a more thorough characterization of the phase separated Ede1 structures observed in mutant cells or after over-expression. However, we are still not completely convinced phase separation does happen during endocytosis and is functionally important for this process. We agree with the authors that all the data are "compatible" with this idea, but they are also compatible with the idea that phase separation is a protection mechanism for excess protein or non-functional proteins. The authors acknowledge that in the response to reviewers ("we cannot exclude the possibility that the condensation of some constructs in the wild-type background is due to the truncations or the increase in protein levels") and use "likely" very often in their response when talking about phase separation during endocytosis. Most arguments we used in our first review are still valid even with the new data. We acknowledge that completely disproving the alternative hypotheses will be tedious but at least the text should present more clearly that several interpretations are compatible with the data, and the title and abstract should be toned down.

2. The new data showing Ede1 concentration buffering is interesting. However, we find the authors' interpretation unclear at places and, to our understanding, only one possible interpretation of the data. The authors say the cells control their cytoplasmic concentration of Ede1 but, to them, is it an active or a passive process? For bona fide purely phase separating proteins, this can be a passive process (once phase separation happens the soluble concentration remains virtually constant and the size of the condensates changes with changing concentration). The data presented in the paper is indeed compatible with this hypothesis. However, the data is also compatible with equally likely alternative hypotheses. One is that there are two populations of Ede1 with different post-translational modifications because of the overexpression. As the authors point out, Ede1 is the most heavily phosphorylated protein in CME (Lu et al. 2016), and it is also ubiquitinated. Therefore, it is possible that overexpressing Ede1 overloads its kinases, phosphatases and/or ubiquitination machineries, and the excess mis-phosphorylated/mi-ubiquitinated proteins would phase separate, without being involved in endocytosis. It is also possible that Ede1 does not normally phase separate at endocytosis sites because it is bound to endocytic partners or cargo. Overexpressing Ede1 would create a stoichiometric imbalance at endocytic sites and the excess protein would end up in the cytoplasm and could phase separate there only.

3. The new data where the PQCC domains are replaced with known phase separating domains is nice and interesting. We are wondering if phase separation is required or if strong dimerization or oligomerization would be sufficient for the rescue? This hypothesis is compatible with the fact that the effects of the ΔCC and ΔPQ deletions seem additive and that the ΔCC (possibly a dimerization domain) mutant has very weak localization to endocytic patches. To test this hypothesis, the authors could try to rescue the ΔCC, ΔPQ and ΔPQCC mutants with dimerization or oligomerization domains, instead of phase separating domains.

---

## [Author Response]

[Editors’ note: the authors resubmitted a revised version of the paper for consideration. What follows is the authors’ response to the first round of review.]

In summary, there is a consensus on the importance of the scientific question addressed by your work, and on the nature of the Ede1 condensates when it is overexpressed or in the endocytic adaptor mutant background. However, 2 of 3 reviewers considered that the manuscript does not demonstrates that Ede1 really undergoes liquid-liquid phase separation at endocytic sites under physiological conditions and whether this property is required for its endocytic function. The concerns follow.

We now describe several of the suggested experiments, as well as some which were not suggested but, in our view, significantly strengthen the manuscript. We also rewrote the Discussion section to more effectively ground our findings in the broader context of endocytic assembly and the evolving understanding of phase separation processes.

1. Several of the key experiments aimed at showing phase separation yield different results for the condensates and the endocytic sites. Basically, they disassemble at different temperatures and at different hexanediol concentrations. Together with the fact that FRAP per se does not demonstrate that Ede1 forms a separate liquid phase at endocytic sites, the data rather suggest that the Ede1 condensates and the Ede1 endocytic patches have different properties.

These are valid concerns, but we do not see these results as contradictory. We do not wish to imply that phase separation is the sole driver of endocytosis, but rather, that Ede1 promotes endocytosis through a mechanism consistent with the phase separation framework. The machinery found at *bona fide* endocytic sites is of course complex and contains many types of interactions, not all of which can be found in abnormal Ede1 condensates. Importantly, we now show that, under normal wild type conditions, cells maintain the cytosolic Ede1 level at the critical concentration for phase separation (new Figure 3). Therefore, even in normal cells Ede1 is poised to undergo phase separation.

We rewrote several portions of the text, including most of the Discussion, to convey our reasoning with more clarity and to highlight how assembly via phase separation is distinct from other mechanisms of assembly.

2. The experiment showing that domains required to form the condensates are required for endocytosis is not really conclusive because, among other considerations, there seems to be an inverse correlation between the capacity of the different mutants to be recruited at endocytic sites and their capacity to sustain endocytosis. Under these circumstances, the endocytic defect observed in the ∆PQCC mutant could be due to the absence of the N or the C-terminal portions of Ede1 at endocytic sites, rather than the absence of polyQ and coiled-coil domains.

We agree that the terminal regions of Ede1 are also important for endocytosis. After all, the central region alone is also not efficiently recruited to endocytic sites, forming instead intracellular condensates.

However, we do not see this as a contradiction. To the best of our knowledge, the N- and C- terminal regions harbour interaction motifs and not catalytic domains. In our model, the primary function of Ede1 is to concentrate itself (through phase separation of the self-interacting central region) and other proteins (through the N- and C-terminal interaction domains). If Ede1 cannot form patches at the plasma membrane as a consequence of ∆PQCC mutation, it cannot perform this function.

Nevertheless, we performed the experiment suggested below, and found that artificial recruitment of the ∆PQCC mutant to other endocytic proteins cannot rescue its function.

Demonstrating conclusively that Ede1 undergoes a phase transition at endocytic sites and this property is required for endocytic uptake is challenging and will take considerable effort and time. But in essence, these are the points that would actually move the field forward, since phase transitions have been now demonstrated for many proteins, but its functional relevance in many contexts including endocytosis, remains elusive.We list here a few experiments that could help you prepare a stronger manuscript, but there are likely many more.(1) to hook the wild type Ede1 or the ∆PQCC mutant to an early endocytic protein (i.e. Syp1) in an ede1∆ background and show that the strain expressing the mutant but not the wild type still has an endocytic defect;

We have now performed this experiment using the anchor-away inducible dimerization technique with Ede1 and Syp1. Neither the localization of the Ede1 mutant nor Sla1 patch density change significantly after rapamycin addition (Figure 6—figure supplement 1). The recruitment of Ede1 EH domains and C-terminus to membrane-binding endocytic proteins is thus insufficient to promote endocytosis on its own.

(2) to finely titrate the concentration of cytosolic Ede1 from 0 to physiological levels and show that the concentration of Ede1 at endocytic sites is sigmoidal rather than linear;

We believe that the experiment suggested here, while potentially very valuable, would pose significant technical challenges, from titrating the in vivo concentration finely enough to imaging the earliest stages of site formation with sufficient sensitivity to support this kind of analysis. Furthermore, we don’t think that the concentration of Ede1 at the endocytic sites is solely determined by a phase separation like process, but also via more “classical” interactions with the lattice of the endocytic coat.

This suggestion however, led us to test a related prediction of phase separation, namely the existence of a critical protein concentration in the cytosol (new Figure 3). We measured the cytosolic and total cellular fluorescence intensity of tagged Ede1 and found that the cytosolic concentration does not increase above wild-type levels even in cells in which Ede1 is over-expressed. This suggests that the normal cytosolic concentration of Ede1 is buffered by phase separation, and that the excess Ede1 is condensed at endocytic sites, and at higher expression levels into the Ede1 condensates. The fact that cells maintain cytosolic Ede1 at the critical concentration means that the phase separation activity of Ede1 is highly likely to contribute to its normal assembly at the endocytic sites.

(3) to show that another IDR capable of undergoing liquid-phase separation can substitute for the PQ and CC domains of Ede1; or

We substituted four different low-complexity domains of different lengths for the central region of Ede1 (new Figure 8). The domains were chosen based on criteria such as prion-like sequence and known tendency to phase separate. The replacements partially rescue the localization and function of Ede1. Although we couldn’t fully re-engineer the complete functionality, we think this provides support for phase separation activity contributing to Ede1’s normal function.

(4) to demonstrate that the PH-PQ-CC construct can recruit the actin machinery and trigger actual endocytic events, but that this capacity is lost under experimental conditions that dissolve the Ede1 condensates (5 min 37oC or 5% hexanediol).

The experiment presented in Figure 9 (former Figure 7) was meant to merely illustrate the ability of Ede1’s central region to concentrate membrane-bound proteins. Finding that it interacts with the endocytic machinery was unexpected, and we think that the full characterization of this interaction is beyond the scope of this article.

Reviewer #1:The manuscript addresses what has been a debated issue in the last years regarding the role of intrinsic disordered domains and liquid-liquid phase separation in endocytosis. While it is clear that many endocytic proteins including for example AP180 or Epsins contain this kind of domains, demonstration of a role of liquid-liquid phase separation in the context of membrane budding in vivo has remained elusive. In this work, Kozak and Kaksonen identify a region in Ede1, an early endocytic factor of the yeast *S. cerevisiae*, which seems to be necessary and sufficient to trigger liquid-liquid phase separation when overexpressed in vivo (based on 4 criteria: the observation of fission and fusion events, fast recovery upon photobleaching, reversible disassembly upon heating and sensitivity to 1,6-hexanediol). Deletion of this region in Ede1 causes endocytic defects, whereas its covalent attachment to a PH domain, which targets it to the plasma membrane, generates cortical patches that recruit other endocytic components and seem to trigger endocytic events.While the data is interesting, it is still a bit preliminary, and it does not really probe at the moment that the putative liquid-liquid-phase separation induced by this Ede1 region is the relevant feature required for endocytic uptake. The authors would need to take into consideration a few points:The authors show that the Ede1 foci undergo fusion and fission events and use this observation to support that the Ede1 foci correspond to a separated liquid phase. How often are these events observed? Are those fusion and fission events associated to membranes?

We did not quantify the exact frequency of fission or fusion events as these are regular, but relatively rare. A rough estimate would be that in a field of a few dozen cells, we will usually observe one such event per minute.

As the condensates are frequently associated with membranes, the fission and fusion events are as well, but we have also seen free-floating condensates split and/or fuse back.

Does Ede1 undergo liquid-liquid phase separation in vitro? The authors mention that this is the case for the mammalian homolog Eps15, but they would need to show that this is the case for Ede1.

We agree that in vitro experiments would provide valuable insights. However, we believe that we have provided a substantial amount of in vivo evidence. Establishing a protein purification pipeline would significantly delay publication of the in vivo data.

Other endocytic proteins have intrinsically disordered domains. How is temperature and 1,6-hexanediol affecting these proteins. Could the authors include controls of other bona-fide non-membrane bound organelles for comparison?

The effects of 1,6-hexanediol indeed apply to a wide range of membraneless organelles (Kroschwald et al., 2015; Molliex et al., 2015; Nott et al., 2015). We chose to focus on other suggested experiments rather than repeat the hexanediol treatments found in literature. We discuss the non-specific nature of hexanediol treatment in the manuscript and note that it is only one of several lines of evidence.

Other authors have shown that late coat proteins like Sla1 are also susceptible to dissolution by temperature, but at higher values than Ede1 (Bergeron-Sandoval et al., 2017). Our experience with Sla1 confirms that it is more stable than Ede1. A full screen for the sensitivity of endocytic proteins to temperature treatments would be an interesting idea but outside of the scope of this manuscript.

The authors show that deletion of the Ede1 region inducing liquid-liquid phase separation install endocytic defects, but the corresponding construct is not recruited to endocytic sites, so the defects could also be caused by the absence of other Ede1 domains. The authors would need to artificially hook the Ede1 mutant to another early component (i.e. Syp1) and check then if an endocytic defect is installed.

We have now performed the suggested experiment by attaching Ede1∆PQCC-mCherry-FRB to Syp1-FKBP via rapamycin-induced dimerization. We did not observe a rescue of the patchy Ede1 localization pattern. We also quantified the number of Sla1-EGFP sites with and without rapamycin and detected no statistically significant increase in Sla1 density (Figure 6—figure supplement 1). We conclude that the mere recruitment of Ede1 N- and C-terminal regions to other membrane interacting endocytic proteins is insufficient to promote the initiation of endocytosis.

Are the effects of the PH-Ede1 construct, and in particular Sla1 recruitment, especially sensitive to temperature and/or 1,6-hexanediol treatment?

We abandoned the idea of using 1,6-hexanediol on the Ede1PQCC-2xPH construct during preliminary experiments. We found that hexanediol treatment causes the PH domain itself to form clusters, or possibly membrane invaginations. This finding is another demonstration of the wide-ranging effects of hexanediol on living cells. While we think it is only peripherally related to the topic of the article and thus did not include it, we can add the data if requested.

To address this question, we have however added a FRAP experiment which shows that the structures formed by the Ede1PQCC-2xPH construct have a solid-like, rather than liquid-like, state (Figure 9—figure supplement 1). We discuss this observation on line 483 of the manuscript: “However, the GFP-Ede1PQCC-2×PH puncta persist over long imaging periods and do not disassemble after Sla1 internalization. Structures formed by GFP-Ede1PQCC-2×PH also do not recover fluorescence. This suggests that while the central region of Ede1 can drive phase separation, the terminal regions are needed to maintain the liquid state.”

It is difficult to evaluate from the kymographs in figure 7 if Sla1 fades away or it is internalized. Can the authors show images of wild type Ede1/Sla1 patches and quantify internalization in both situations? Is Abp1 also recruited to the PH-Ede1 foci? This would be expected if internalization occurs.

The kymographs in Figure 9 (formerly Figure 7) were generated from TIRF movies (Figure 9—video 1) and therefore cannot show inward movement, which happens along the optical axis. However, the equatorial plane movie shows inward movement of Sla1 patches (Figure 9—video 2).

Reviewer #2:Kozak and Kaksonen propose that Ede1 may initiate endocytic events due to its phase separation properties. We found this manuscript very interesting. There's no escaping phase separation right now in cell biology, but this manuscript strikes a nice balance between hype and reality because they are upfront about what is studied here (by overexpression or expression in mutant background) and what it suggests may be happening in normal cells during endocytosis. We think the experiments presented do support the conclusions. Complementary work on bioRxiv shows similar behaviour of the mammalian homologue Eps15, which gives confidence that the authors are on the right track here.There was one point that if addressed experimentally (under non-pandemic circumstances) would strengthen the manuscript. The data on the condensates are convincing, but we wondered whether the Ede1 blobs are truly liquid condensates that are devoid of membrane. In our own work in mammalian cells we have seen large blobs of overexpressed protein that turn out to be MVB-like structures by EM. These structures show similar FRAP profiles to the data here, and I don't think any of the experimerts presented here necessarily rule out that these structures may have a membrane component.

This is a reasonable hypothesis that we had already planned on addressing with electron microscopy. However, during the course of preparing and revising the manuscript, other authors published independent, complementary work investigating the removal of endocytic protein condensates via Ede1-mediated autophagy (Wilfling et al., 2020). They performed correlated cryo-ET experiments on the Ede1 droplets in overexpression yeast strains and confirmed that there was no vesicle component in the condensates. Rather, the endocytic condensate appears as a large ribosome exclusion zone comparable to that seen around normal endocytic sites (Kukulski et al., 2012). Wilfling et al. also noted that the endocytic condensates were frequently contacted, but not completely enclosed by, ribosome-containing tubular membranes assumed to be the ER.

Our remaining points are straightforward to address with existing data and would help to improve the paper.1. Figure 1D number of cells with condensates is annotated. Are there differences in the number or brightness of condensates between EDE1-EGFP/EDE1-EGFP and EDE1-EGFP/EDE1?

As expected, heterozygous cells with only one tagged Ede1 allele are on the whole dimmer than the homozygotes, as both the sites and the condensates in heterozygous cells contain some amount of tagged and untagged Ede1. We did not notice any difference in number.

2. Figure 2C would be improved by including the quantification that is listed in the text i.e. graph with average and fit.

We thank the reviewer for the suggestion. We provided this quantification in panel 2D.

3. Figure 3 who recruits who in this figure? It is convincing that the blob is due to Ede1 but the site of the blob could be dictated by any one of the other proteins. Do movies of blob formation show that each protein appears after Ede1 or do they simultaneously accumulate?

Because the cells are genomic knockouts (triple deletion) or constitutive overexpression, and the condensates persist throughout the cell cycle (Figure 1—video 1), we do not actually have movies of de novo condensate formation.

An exception is the new experiment with galactose-induced Ede1 overexpression in new Figure 3 (see Figure 3—video 1 in particular), but in these cells, only Ede1 was fluorescently tagged. The condensates in this experiment do initially appear as normal endocytic sites but end up growing instead of progressing into movement and disassembly.

4. Figure 4C shows FL gives 100% of cells with condensates but the percentage is lower in Figure 1D – is this just experimental variability?

We added more quantifications to provide a better idea of the variability present in these experiments. The strains in question are not the same (Figure 1D is diploid, 4C is haploid) but from the data we have, it does not appear they are significantly different in terms of the fraction of cells with Ede1 condensates.

5. Figure 5CD is there a correlation between patch density and lifetime? If they are plotted against each other do they scale linearly? Also, the movies in the manuscript work well and for this figure, a side-by-side movie of wt, ∆PQCC and ede1∆ would be a useful addition.

The plot in Author response image 1 shows mean lifetime and density values and their 95% confidence interval (based on the inter-experimental variability). There is a fairly robust correlation (adjusted R2 = 0.93), shown is a trend line with 95% confidence interval (shaded area).

**Author response image 1. sa2fig1:** 

We carefully considered adding more movies to the manuscript. In this case, the phenotype is subtle and requires careful quantification. We believe that the movies on their own would not be illustrative enough to the readers to warrant their inclusion.

6. Figure 6 shows a representative cell. Quantification would strengthen this figure.

We completely agree with the reviewer. However, the phenotype presented in Figure 7 (formerly Figure 6) is one of a reduced tendency of early adaptors to cluster – ‘spotty’ vs ‘fuzzy’ localization at the sites. This is unlike the phenotype presented in Figure 6 (formerly Figure 5), where Sla1 sites are reduced in number, but retain good contrast for thresholding and counting. In addition, the phenotypes are similar, but not the same for all proteins. For example, Syp1 loses the punctate localization on the plasma membrane, but remains bright at the bud necks, where it localizes independently of Ede1.

We were thus unable to find a measure of signal dispersion that would be meaningful and generalizable to all strains presented in this figure, and chose to present the result in a qualitative manner.

Reviewer #3:In this paper, Kozak and Kaksonen report that the budding yeast's early endocytic protein Ede1, a homologue of mammalian Esp15 can form aggregates when the protein is overexpressed or expressed in absence of 3 other early endocytic proteins (the AP180 family proteins Yap1801/2 and the AP2 complex α-subunit Apl3). The paper aims at showing that these aggregates are phase separated structures and speculate that phase separation could be the mode of assembly of Ede1 during clathrin-mediated endocytosis in yeast. The authors also identified the minimum portion of Ede1 that leads to these aggregates.1. The data are relatively solid to demonstrate that the aggregates seen in the mutants can be phase separated structures. However, I am not convinced the data strongly support that Ede1 experiences phase separation at sites of clathrin-mediated endocytosis. To my opinion, several pieces of data strongly suggest that Ede1 assembles into two different types of structures in the mutant strains – one structure at the endocytic sites, which is due to specific high affinity interactions, and one structure at the aggregates that is partly due to new low affinity interactions that are not present in normal conditions at the endocytic sites (possibly in addition to the other high affinity interactions mentioned above). Indeed, several of the key experiments that aim to show phase separation yield different results for the aggregates and the endocytic sites:(1a) Aggregates and endocytic sites are disassembled at different temperature. This is expected from the laws of thermodynamics if the interactions in aggregates and endocytic sites have different affinities. Indeed, the Kd of any reversible biochemical interaction depends exponentially on the temperature and low affinity interactions (from the aggregates) will become unfavorable at lower temperature than higher affinity interactions (from the endocytic sites).

Ede1 has many different known interactions: both self-interaction via its core region and several interactions with various other endocytic proteins via its N- and C-terminal domains. It is highly likely that these interaction networks are not identical between endocytic sites and the condensates leading to different affinities. We don’t think that the different affinities are contradicting our conclusions.

(1b) 1,6-hexanediol dissociates aggregates at lower concentrations than it dissociates endocytic patches, which also suggest the interactions in the aggregates have lower affinity. As a side note, I want to point out that even though researcher in the phase separation field routinely use hexanediol to demonstrate phase separation, it remains somewhat controversial [McSwiggen et al. 2019].

We completely agree with the reviewer as to the controversial nature of 1,6-hexanediol. It is known to have unexpected systemic side-effects (Kroschwald et al., 2017) and long treatments can even induce stress granule formation (Wheeler et al., 2016). We are aware of these limitations, which is why (a) the duration of our experiments with hexanediol was kept to a minimum, and (b) it is only one of several applied criteria. We now state this explicitly in the text.

(1c) FRAP on endocytic sites does not demonstrate that they are phase separated structures: many other endocytic proteins do exchange rapidly and are not believed to be phase separated (e.g. the actin cytoskeleton proteins [Kaksonen et al. 2003, Kaksonen et al. 2005, Lacy et al. 2019]).

We agree again with the reviewer: our FRAP data at endocytic sites is consistent with phase separation but does not conclusively prove it. We now discuss the technical challenges of testing phase separation at endocytic sites in the Discussion section. However, we believe that in this revised manuscript we have provided data to argue that phase separation of Ede1 is highly likely to take place at the endocytic sites. Particularly, we now show that the cytosolic concentration of Ede1 in wild type cells is maintained at the critical concentration for phase separation. Therefore, it is expected that any additional Ede1, which in normal cells is accumulated at endocytic sites, is poised to phase separate.

(1d) The mobile and immobile fractions are different in aggregates and endocytic sites.

Phase separated condensates are known to exist in different states from highly fluid to gel-like to solid. It is conceivable that the fluidity of Ede1 condensates is modulated by different cellular activities such as different interactions, or post-translational modifications such as phosphorylation or ubiquitination.

2. Since Ede1 concentration seems to be critical for the formation of the aggregates, it is important to systematically report the cytosolic and cell concentrations (or even the relative intensities) in different experimental conditions:(2a) l. 100, it is suggested that the levels of cytoplasmic Ede1 might be different in wild-type and the triple mutant. Concentrations (even relative) should be reported.

We thank the reviewer for this excellent suggestion. We now quantify the cytosolic and total cell concentrations (new Figure 3). We found that there is a limit to the cytosolic concentration of Ede1, which is in line with the predictions made by a model of phase separation driven primarily by homotypic interactions. Importantly, this limit is reached by wild-type cells, suggesting that Ede1 phase separates in normal conditions.

We also show that the total Ede1 levels are mildly increased in the triple deletion mutant (Figure 3, Figure 5—figure supplement 1).

(2b) In Figure 4 it is important to make sure the aggregation with different constructs is not due to differences in the cytoplasmic or cell concentrations of the constructs.

We have now addressed this question by western blotting (Figure 5—figure supplement 1). All of the concentrations of truncated constructs are elevated compared to the full-length in wild-type background cells.

At this point we cannot exclude the possibility that the condensation of some constructs in the wild-type background is due to the truncations or the increase in protein levels.

On the other hand, the lack of condensates in the triple deletion mutant in constructs lacking the PQ region or the coiled-coil cannot be explained by reduced protein levels. This is particularly true for 590-1381 and 1-590 variants which are present at elevated concentrations and do not form condensates. Therefore, we conclude that the diffuse localization of these constructs is due to molecular features and not concentration.

(2c) Figure 7C and l. 252+: how the behavior of the puncta formed with the Ede1(366-900)-2xPH construct depend on concentration? Also, do all the puncta recruit Sla1? Is Sla1 also recruited at sites where the Ede1 construct is not present?

The size and shape of structures formed by the Ede1(366-900)-2xPH construct depend on the concentration of the construct (Figure 9A,B; formerly 7A,B). Low expressing cells had more diffuse localization of the construct and separated into puncta or well-separated regions as the concentration increased.

Most of the puncta of Ede1 construct do recruit Sla1, but Sla1 is also recruited at sites where the Ede1 construct is not present (see kymograph in former Figure 7C, now 9F). The artificial Ede1 construct is likely less effective in recruiting the endocytic components than the natural Ede1. Thereby, allowing some sites to form independently of Ede1 as is seen in the Ede1 deletion strains.

3. l. 157: I am not sure I understand the logic (I may have missed something). The fact that it is a condensate does not imply it triggers actin polymerization. This is a correlation, not a causation. In addition, one could imagine there is a defined endocytic patch within a diffraction limited distance from the condensate. Quantifying the amount of abp1 (relatively to other patches) could be a way to tell whether these are bona fide endocytic structures coming out of the aggregate.

We did not mean to imply that actin polymerization is directly caused by the phase-separated nature of the condensates, merely that proteins present in the condensates are normally able to trigger it (e.g., Las17). We edited that line for clarity (l. 202).

4. In conclusion, in its current form, this study does convince me that Ede1 phase separates during clathrin-mediated endocytosis. In addition, it is unclear to me how the putative phase separation of Ede1 would affect endocytosis, compared to regular protein assembly. I believe several extra experiments would be necessary to make a strong point.

We thank the reviewer for their critical assessment. We hope that the new experimental data we provide in this revised manuscript now convinces the reviewer of the value of our findings. Particularly, the two key additions are: we show that the normal cytosolic Ede1 level is at the critical concentration for phase separation (new Figure 3), and that the phase separating core region of Ede1 can be functionally replaced with heterologous sequences that are known or predicted to phase separate (new Figure 8).

It is true that phase separation of diffraction-limited structures is a contentious topic that requires critical assessment (McSwiggen et al., 2019; Peng and Weber, 2019). Our new findings add to the lines of evidence already present in the original manuscript, which all point to the same phenomenon. We rewrote the discussion to put our findings in the broader context and distinguish phase separation from other types of assembly.

References

Bergeron-Sandoval, L.-P., Heris, H. K., Hendricks, A. G., Ehrlicher, A. J., François, P., Pappu, R. V., & Michnick, S. W. (2017). Endocytosis caused by liquid-liquid phase separation of proteins. BioRxiv, 145664. https://doi.org/10.1101/145664

Kroschwald, S., Maharana, S., Mateju, D., Malinovska, L., Nüske, E., Poser, I., Richter, D., & Alberti, S. (2015). Promiscuous interactions and protein disaggregases determine the material state of stress-inducible RNP granules. ELife, 4, e06807. https://doi.org/10.7554/eLife.06807

Kroschwald, S., Maharana, S., & Simon, A. (2017). Hexanediol: A chemical probe to investigate the material properties of membrane-less compartments. Matters, 3(5), e201702000010. https://doi.org/10.19185/matters.201702000010

Kukulski, W., Schorb, M., Kaksonen, M., & Briggs, J. A. G. (2012). Plasma membrane reshaping during endocytosis is revealed by time-resolved electron tomography. Cell, 150(3), 508–520. https://doi.org/10.1016/j.cell.2012.05.046

McSwiggen, D. T., Mir, M., Darzacq, X., & Tjian, R. (2019). Evaluating phase separation in live cells: Diagnosis, caveats, and functional consequences. Genes & Development, 33(23–24), 1619–1634. https://doi.org/10.1101/gad.331520.119

Molliex, A., Temirov, J., Lee, J., Coughlin, M., Kanagaraj, A. P., Kim, H. J., Mittag, T., & Taylor, J. P. (2015). Phase separation by low complexity domains promotes stress granule assembly and drives pathological fibrillization. Cell, 163(1), 123–133. https://doi.org/10.1016/j.cell.2015.09.015

Nott, T. J., Petsalaki, E., Farber, P., Jervis, D., Fussner, E., Plochowietz, A., Craggs, T. D., Bazett-Jones, D. P., Pawson, T., Forman-Kay, J. D., & Baldwin, A. J. (2015). Phase transition of a disordered nuage protein generates environmentally responsive membraneless organelles. Molecular Cell, 57(5), 936–947. https://doi.org/10.1016/j.molcel.2015.01.013

Peng, A., & Weber, S. C. (2019). Evidence for and against liquid-liquid phase separation in the nucleus. Non-Coding RNA, 5(4), 50. https://doi.org/10.3390/ncrna5040050

Wheeler, J. R., Matheny, T., Jain, S., Abrisch, R., & Parker, R. (2016). Distinct stages in stress granule assembly and disassembly. ELife, 5, e18413. https://doi.org/10.7554/eLife.18413

Wilfling, F., Lee, C.-W., Erdmann, P. S., Zheng, Y., Sherpa, D., Jentsch, S., Pfander, B., Schulman, B. A., & Baumeister, W. (2020). A selective autophagy pathway for phase-separated endocytic protein deposits. Molecular Cell. https://doi.org/10.1016/j.molcel.2020.10.030

[Editors’ note: what follows is the authors’ response to the second round of review.]

After some discussion on the new version of the manuscript, there is a general agreement on the interest of the article and the quality of the data. However, two out of three reviewers still consider that the functional relevance of liquid phase separation in endocytosis is not conclusively proven and other hypothesis might as well explain the data. Even though the observation that other unrelated Intrinsically Disorder Regions (IDRs) can partially substitute for the Ede1 regions promoting liquid phase separation is quite compelling, the inserted IDRs might as well work by providing a flexible loop of a given size between the N and C terminal domains or by promoting oligomerization. Since the role of liquid phase separation in endocytosis is under strong debate, we feel that these two other possibilities need to be properly tested and discussed before publication. Even if the new data would disprove the liquid phase separation model, the article would still be of interest for eLife. If the new oligomerization or linker regions cannot functionally substitute for the Ede1 coiled-coil (CC) and PQ rich (PQ) regions, the data would provide stronger support for the role of liquid phase separation in endocytosis but still, the title, the abstract and the discussion would need to be rephrased to avoid overstatements that would slow down advance in the field. In addition to this main concern, there is still the issue on the recruitment of the Ede1-(PQ,CC)∆ mutant to endocytic sites. At this point, it is not possible to conclude that the CC-PQ region is required for endocytic uptake because the mutant is not recruited to endocytic sites. One of the reviewers also suggested to investigate the possible differences in the posttranslational modifications that could lead to phase-separation. Even though this approach could be enlightening, it will take more than two months and it is probably beyond the scope of the manuscript. In essence then, the manuscript would be adequate for publication in eLife, provided than you address the following key points:1. Substitute the CC and PQ regions for non-IDRs that can act as linkers or oligomerization domains and investigate if the chimeras can complement the ede1∆ phenotype.

We expanded the repertoire of protein linkers investigated in Figure 8 with four structured domains: two fluorescent proteins serving as globular linkers, and two kinesin coiled-coils. In addition, these linkers run the gamut of oligomerization states, from monomeric (mCherry), through dimeric (dTomato, kinesin-1) to tetrameric (kinesin-5).

The results suggest that Ede1 localization, but not function, can be partially rescued by an oligomerization domain. However, we saw a significant functional rescue of the late-phase patch density only when we used prion-like IDRs as replacements.

These results are consistent with the model where EH domains mediate interactions with other proteins, and the central region mediates self-interaction and condensation. Thus, increasing the valency of EH domains can rescue the recruitment of Ede1 to functional patches, but it did not rescue Ede1 function of promoting patch formation.

We would like to note that as a general principle, it is difficult to interpret this type of in vivo domain replacement experiment regardless of the result. We performed and then expanded this experiment per the reviewers’ requests, but we try to approach the results presented in Figure 8 with caution. In living cells, there are too many uncontrollable variables for us to be able to say conclusively why one replacement can rescue the defect and the others cannot. Still, these results do support our conclusion that heterologous prion-like IDRs can, at least partially, replace the function of Ede1’s core region.

2. Solve the issue of the recruitment of the Ede1-(PQ,CC)∆ to endocytic sites.

We showed previously that the deletion of the central region of Ede1 inhibits the formation of endocytic patches by the early-arriving endocytic proteins (Figure 7) and that this has downstream effects on the late-arriving Sla1 protein (Figure 6).

The reviewer suggested a possibility that the central region of Ede1 might be simply a recruitment motif, and that the terminal fragments of Ede1 could support early site formation if recruited to the membrane or endocytic proteins through other means.

We tested that possibility in the previous round of revision by artificially recruiting the Ede1 terminal domains to the early protein Syp1. This, however, did not rescue the initiation defect nor the downstream defect.

We cannot help but see this reasoning as somewhat circular: because the early patch formation requires Ede1 central region (Figure 7), we cannot recruit Ede1 which lacks the central region to early patches. We can only recruit it to other membrane-binding early endocytic proteins, which we did for Syp1, showing no rescue of Syp1 or Ede1 patch localization, and no effects downstream on the late phase proteins.

We have now repeated this experiment by recruiting the Ede1 terminal domains to Sla2 (Author response image 2). We did not feel that these data would add to the conclusions of the manuscript, so we decided to only show them here in the rebuttal.

We chose Sla2 as an anchor, because the localization of Sla2 to endocytic sites is not prevented by the Ede1 central region deletion (Figure 7). Recruiting Ede1 terminal domains to Sla2 did rescue Ede1 patch formation, although not fully (Author response image panel A). However, this did not rescue the downstream Sla1 patch density (panel B, patch density from respectively 47 and 69 cells in DMSO- and rapamycin-treated cells; line and whiskers show mean +/- SD; p-value from two-sided t-test).

As a control for the rapamycin-driven recruitment, we used a strain described by Brach et al. (2014), where three early proteins tagged with FRB are recruited to eisosomes via FKBP-tagged Pil1, and subsequently recruit Ede1-EGFP. Rapamycin treatment of this strain under the same conditions as the Sla2/Ede1∆PQCC strain resulted in increased Ede1-Pil1 colocalization compared to solvent-only treatment (panel C).

These results agree with our conclusion that the central region of Ede1 is important for its function. However, we do not feel that negative results from this kind of synthetic biology experiments are very conclusive. Alternative explanations could be that we did not recruit the correct amount of Ede1 terminal domains, or that they are not correctly localized or oriented with respect to the binding sites in the other early endocytic proteins.

In these experiments we are attempting to re-engineer Ede1, a relatively complex multidomain protein, by replacing a part of it with a heterologous sequence. These kinds of experiments are obviously very challenging, and the likelihood of failure is high. In the manuscript we conclude that the core region of Ede1 is critical for assembling Ede1 and other early endocytic proteins to endocytic sites. Furthermore, we are suggesting that the condensing ability of the core region is important for Ede1’s function. Even if we were able to engineer a variant of Ede1 that functions without self-association driven condensation, our conclusions about the function of normal Ede1 would not be affected. We never suggested that the condensing property of Ede1 would be an absolute requirement for endocytosis, nor that it could not be replaced with another mechanism. Therefore, we believe that further attempts at making Ede1 localize functionally without its natural core region are outside of the scope of this manuscript.

3. Rewrite the title, the abstract and the discussion to make clear that the results support the requirement for liquid-phase separation in endocytosis but they do not conclusively prove it.

We have changed the title to use the mechanism-agnostic term *Condensation* in place of *Phase separation*. We have also made numerous changes to the text. On the whole, we consider the text to have been fairly cautious and account for possible alternative hypotheses already. Reviewer #2 notes: “*(…) I think this is a careful study and the authors are quite conservative in their interpretation.*”

In addition to these main concerns, please carefully read the rest of the points raised by the reviewers and discuss or address them:1. The new data showing Ede1 concentration buffering is interesting. However, we find the authors' interpretation unclear at places and, to our understanding, only one possible interpretation of the data. The authors say the cells control their cytoplasmic concentration of Ede1 but, to them, is it an active or a passive process? For bona fide purely phase separating proteins, this can be a passive process (once phase separation happens, the soluble concentration remains virtually constant and the size of the condensates changes with changing concentration). The data presented in the paper is indeed compatible with this hypothesis. However, the data is also compatible with equally likely alternative hypotheses. One is that there are two populations of Ede1 with different post-translational modifications because of the overexpression. As the authors point out, Ede1 is the most heavily phosphorylated protein in CME (Lu et al. 2016), and it is also ubiquitinated. Therefore, it is possible that overexpressing Ede1 overloads its kinases, phosphatases and/or ubiquitination machineries, and the excess mis-phosphorylated/mi-ubiquitinated proteins would phase separate, without being involved in endocytosis. It is also possible that Ede1 does not normally phase separate at endocytosis sites because it is bound to endocytic partners or cargo. Overexpressing Ede1 would create a stoichiometric imbalance at endocytic sites and the excess protein would end up in the cytoplasm and could phase separate there only.

We completely agree that condensation properties of Ede1 may be affected or regulated by phosphorylation and / or ubiquitylation. This regulation deserves investigation but is outside the scope of this article. We mention Ede1 phosphorylation in the Discussion section of the manuscript (p. 22).

On this point, we would also like to stress that one of the conclusions from the quantifications presented in Figure 3 is not merely that a threshold cytoplasmic concentration of Ede1 exists, but that wild-type cells already operate at this threshold level.

2. We find it surprising that the FUS domain does not phase separate Ede1 since FUS is a well known phase separating domain, and other phase separating domains seem to work. Do the authors have an explanation for that?

We can offer plausible speculation. FUS family proteins phase separate through a distinct mechanism mediated by tyrosine and arginine residues, and affected by phosphorylation and methylation (Qamar et al., 2018; Wang et al., 2018). The FUS low-complexity (LC) region we used readily phase separates at high concentrations in vitro*,* but in physiological conditions, FUS phase separation is greatly enhanced by the interactions of the LC region with the RNA recognition motif (Wang et al., 2018). Previous work in yeast has shown that FUS aggregates with cytotoxic effects when overexpressed in yeast (Fushimi et al., 2011). On the other hand, expression of LC-only constructs in yeast reduces the aggregation and cytotoxicity compared to full-length FUS (Kryndushkin et al., 2011), especially when not overexpressed. All of our constructs were expressed from Ede1 locus under the control of the native promoter, and this level may not be enough for the phase separation of the Ede1-FUS construct in yeast cells.

3. L 208: "catalyzes cycles of assembly and disassembly of actin": it is not clearly demonstrated, so this could reformulated.

We have reformulated the sentence to: ”(…) are associated with cycles of assembly and disassembly of actin”.

4. L 392: "likely": change to "possibly" because this is not demonstrated in the paper (just correlation with abp1).

We changed the phrasing.

5. In response to reviewer #2, the authors point out that hexanediol causes the PH domain to form clusters, but say they are not showing it in the paper because it is only peripherally related to the paper. I agree but this could be useful for the field to report this anyway because there are a lot of controversies about the use of hexanediol in the phase separation field.

We agree that this is an important observation, and we included it as part of Figure 9—figure supplement 1.

6. Talking about fusion and fission of condensates is an overstatement. It might just be the condensates getting closer or separating. Talking about apparent fission or fusion events throughout the text would be more accurate. How often are the putative fission events observed?

We agree with the reviewer. Therefore, the text only mentions *apparent fusion and fission*. From single-plane movies, we estimate that apparent fusion and fission events happen at a similar rate of about 0.0007 (cell*s)^-1^; this number would likely be higher if full cell volumes were considered. Apparent fusion often, but not always, follows apparent fission (i.e. a droplet splits when Abp1 is present, and subsequently reforms, as in Figure 4—video 1). We hope the reviewer finds this information useful in the interpretation and assessment of our manuscript.

7. Is the PQ region alone generating condensates? The authors test the 1-591 Ede1 region containing the EH domains and the PQ region, but not the PQ region alone. The EH domain might somehow inhibit the formation of condensates by the IDR.

We have not tried this. It would be an interesting additional experiment, but we believe it is outside the scope of this current manuscript.

8. Are the condensates co-localizing FM4-64 stainable membranes? If so, the membrane association might provide for the liquid-phase-like behavior.

We performed this experiment. The condensates and membranes stained by FM4-64 do not show any apparent colocalization (Figure 4—figure supplement 1).

9. Does Abp1 assemble on all condensates on the time-lapse movies? Does the transient Abp1 assembly on the condensates explain the low percentage of co-localization? Are condensates hot spots for Abp1 assembly?

Not all condensates recruit Abp1 in our movies, but this is unsurprising given the typical duration of ~2 minutes per movie. The transient assembly seen in the movies is the simplest and most obvious explanation for the low co-localization rates, which were measured by analyzing images of single timepoints. Abp1 can assemble on the same condensates multiple times.

10. Are the GFP-Ede1(366-900) condensates behaving as the GFP-Ede1 condensates in terms of temperature and hexanodiol sensitivity? It is surprising that the GFP- Ede1(366-900)-2xPH condensates do not. If the GFP-Ede1(366-900) condensates behave as separate liquid-phase condensates, similar to those generated by Ede1, it might mean that their membrane association modifies the Ede1-liquid-phase properties. In this context, are the PM associated Ede1 condensates (dome-shaped) behaving similar to those assembled within the cytosol?

We present a qualitative comparison of hexanediol effects on GFP-Ede1(366-900) and GFP- Ede1(366-900)-2xPH in Figure 9—figure supplement 1. In short, the membrane-bound construct is not affected by hexanediol treatment. The GFP-Ede1(366-900) condensates are mostly or completely dissolved by hexanediol (there is some cell-to-cell variability; figure shows the typical scenario).

11. It is still not so clear if the GFP- Ede1(366-900)-2xPH can really initiate bona fide endocytic events. The internalization of the Sla1 patches is not clear in the images provided. If they are bona fide endocytic events, the fact that GFP- Ede1(366-900)-2xPH condensates do not behave as a liquid-phase would suggest that the formation of an Ede1-dependent liquid phase is not really essential for endocytic uptake and that the Ede1 central region rather operates as a linker or by recruiting other endocytic proteins.

First, we want to emphasize that we have never claimed that Ede1 condensation in yeast is essential for endocytic uptake. Rather we are suggesting that Ede1 condensation promotes endocytosis by increasing the initiation rate and concentrating cargo-specific adaptors. Ede1 itself is not required for endocytosis. In fact, none of the early-arriving proteins, including clathrin and the AP2 complex, are individually required for endocytic initiation in yeast. We would like to reiterate that the assembly of the endocytic sites is mechanistically flexible and promoted by many different interactions; this point is made repeatedly in the Discussion section of our manuscript.

The take-away of Figure 9 is that the central region of Ede1 can cause clustering of proteins at the plasma membrane. We included the observation that Sla1 also assembles on GFP- Ede1(366-900)-2xPH clusters because we believe it could be interesting to the readers, even if we cannot fully explain it in the present manuscript. Discussion (p.22): “It must be noted that we do not fully understand the nature of the microdomains formed by the GFP-Ede1-PQCC-2×PH construct. For example, all known interaction motifs of Ede1 are located inside the terminal regions. It is thus unclear how the central region could recruit other endocytic proteins.” We have deleted the references to this observation from the abstract and the introduction.

12. The Rapamycin inducible system might not be the most adequate to bring the Ede1 trucations to endocytic sites because the tor1 mutations used to make the yeast strain resistant to Rapamycin might interfere with endocytic uptake. Are Syp1 and Sla1 cortical patch dynamics similar to the wild type´s in a tor1-1 frp1 EDE1-mCherry-FKPB SYP1-FRB background? It might be easier to covalently link the Ede1 truncations to Syp1 or to a later endocytic coat component that is less affected by depletion of Ede1.

The tor1-1 strains have been used previously to study endocytosis and no obvious interference has been observed (e.g. Brach et al., 2014). We agree that there might be other more optimal ways to perform this experiment that was suggested by the reviewer. However, it is very challenging to engineer synthetic protein constructs that could functionally replace normal endocytic proteins.

We re-emphasise the point that we are not claiming that Ede1’s condensation activity is required for endocytic initiation. Rather, we are suggesting that Ede1’s condensation activity contributes to initiation in wild type cells. Therefore, even if we managed to generate a fully functional version of Ede1 that bypasses the need for the natural core region, it would not mean that in normal cells Ede1’s core is not promoting endocytic initiation.

Reviewer #1:Kozak and Kaksonen demonstrate that the mammalian Eps15 yeast homolog Ede1 can phase-separate upon overexpression or deletion of endocytic adaptors in vivo. The authors define the protein regions necessary and sufficient to generate the condensates and demonstrate that those are probably necessary and sufficient to sustain the endocytic function of Ede1. Conclusive demonstration on whether liquid phase-separation actually occurs at endocytic sites in wild type cells remains elusive. However, the authors show that unrelated protein sequences known to undergo this transition can functionally (albeit partially) substitute for the Ede1 region generating the condensates.Even though the data points to the functional role of a separated Ede1 liquid phase in endocytic uptake, some important issues are still unsolved:1. The authors conclude that the coiled-coil and IDR of Ede1 is required for endocytic uptake because using a Rapamycin inducible system to hook the Ede1 mutant lacking these domains to Syp1, does not complement the ede1∆ defects. Again, this particular experiment is not very informative because the Ede1 mutant is not really recruited to endocytic sites in the presence of Rapamycin and therefore, the endocytic defect might be because by the absence of the N or C-terminal domains.

Strictly speaking, neither Ede1 nor any other early-arriving protein, including clathrin, is required for endocytic uptake. We do not claim that they are, and we do propose that multiple mechanisms, including condensation mediated by Ede1, contribute to the assembly of the endocytic coat and the concentration of cargo.

We elaborated on this experiment in the response to the editor. We repeated this experiment by recruiting the Ede1 terminal domains to Sla2. We were successful in partially re-establishing Ede1 patch formation but did not see a change in Sla1 patch density upon rapamycin treatment.

Perhaps the fusion constructs presented in Figure 8 will provide some evidence of the kind requested by the reviewer: the dimeric constructs partially rescue localization but fail to rescue the late-phase site density.

The Rapamycin inducible system might not be the most adequate to bring the Ede1 trucations to endocytic sites because the tor1 mutations used to make the yeast strain resistant to Rapamycin might interfere with endocytic uptake. Are Syp1 and Sla1 cortical patch dynamics similar to the wild type´s in a tor1-1 frp1 EDE1-mCherry-FKPB SYP1-FRB background? It might be easier to covalently link the Ede1 truncations to Syp1.Other important but easily addressable points:1. Talking about fusion and fission of condensates is an overstatement. It might just be the condensates getting closer or separating. Talking about apparent fission or fusion events throughout the text would be more accurate. How often are the putative fission events observed?2. Is the PQ region alone generating condensates? The authors test the 1-591 Ede1 region containing the EH domains and the PQ region, but not the PQ region alone. The EH domain might somehow inhibit the formation of condensates by the IDR.3. Are the condensates co-localizing FM4-64 stainable membranes? If so, the membrane association might provide for the liquid-phase-like behavior.4. Does Abp1 assemble on all condensates on the time-lapse movies? Does the transient Abp1 assembly on the condensates explain the low percentage of co-localization? Are condensates hot spots for Abp1 assembly?5. Are the GFP-Ede1366-900 condensates behaving as the GFP-Ede1 condensates in terms of temperature and hexanodiol sensitivity? It is surprising that the GFP- Ede1366-900-2xPH condensates do not. If the GFP-Ede1366-900 condensates behave as separate liquid-phase condensates, similar to those generated by Ede1, it might mean that their membrane association modifies the Ede1-liquid-phase properties. In this context, are the PM associated Ede1 condensates (dome-shaped) behaving similar to those assembled within the cytosol?6. It is still not so clear if the GFP- Ede1366-900-2xPH can really initiate bona fide endocytic events. The internalization of the Sla1 patches is not clear in the images provided. If they are bona fide endocytic events, the fact that GFP- Ede1366-900-2xPH condensates do not behave as a liquid-phase would suggest that the formation of an Ede1-dependent liquid phase is not really essential for endocytic uptake and that the Ede1 central region rather operates as a linker or by recruiting other endocytic proteins.

We have addressed the reviewer's minor points 1-6 in the response to the editor’s summary.

Reviewer #3:We thank the authors for their efforts in addressing our initial concerns and providing new data. We think the paper has been improved since the first submission. However, we think there are still important remaining issues.1. The authors provide a more thorough characterization of the phase separated Ede1 structures observed in mutant cells or after over-expression. However, we are still not completely convinced phase separation does happen during endocytosis and is functionally important for this process. We agree with the authors that all the data are "compatible" with this idea, but they are also compatible with the idea that phase separation is a protection mechanism for excess protein or non-functional proteins. The authors acknowledge that in the response to reviewers ("we cannot exclude the possibility that the condensation of some constructs in the wild-type background is due to the truncations or the increase in protein levels") and use "likely" very often in their response when talking about phase separation during endocytosis. Most arguments we used in our first review are still valid even with the new data. We acknowledge that completely disproving the alternative hypotheses will be tedious but at least the text should present more clearly that several interpretations are compatible with the data, and the title and abstract should be toned down.

We made several changes to the abstract, as well as changed ‘phase separation’ to ‘condensation’ in the title. We hope that the Discussion section in its current state considers various caveats and alternative mechanisms to the reviewer’s satisfaction.

2. The new data showing Ede1 concentration buffering is interesting. However, we find the authors' interpretation unclear at places and, to our understanding, only one possible interpretation of the data. The authors say the cells control their cytoplasmic concentration of Ede1 but, to them, is it an active or a passive process? For bona fide purely phase separating proteins, this can be a passive process (once phase separation happens the soluble concentration remains virtually constant and the size of the condensates changes with changing concentration). The data presented in the paper is indeed compatible with this hypothesis. However, the data is also compatible with equally likely alternative hypotheses. One is that there are two populations of Ede1 with different post-translational modifications because of the overexpression. As the authors point out, Ede1 is the most heavily phosphorylated protein in CME (Lu et al. 2016), and it is also ubiquitinated. Therefore, it is possible that overexpressing Ede1 overloads its kinases, phosphatases and/or ubiquitination machineries, and the excess mis-phosphorylated/mi-ubiquitinated proteins would phase separate, without being involved in endocytosis. It is also possible that Ede1 does not normally phase separate at endocytosis sites because it is bound to endocytic partners or cargo. Overexpressing Ede1 would create a stoichiometric imbalance at endocytic sites and the excess protein would end up in the cytoplasm and could phase separate there only.3. The new data where the PQCC domains are replaced with known phase separating domains is nice and interesting. We are wondering if phase separation is required or if strong dimerization or oligomerization would be sufficient for the rescue? This hypothesis is compatible with the fact that the effects of the ΔCC and ΔPQ deletions seem additive and that the ΔCC (possibly a dimerization domain) mutant has very weak localization to endocytic patches. To test this hypothesis, the authors could try to rescue the ΔCC, ΔPQ and ΔPQCC mutants with dimerization or oligomerization domains, instead of phase separating domains.

We find reviewer’s points 2-3 useful and have addressed them in the response to the editor’s summary.